# Dynamical fermionization in the one-dimensional Bose-Fermi mixture

Ovidiu I. Pâţu[1]

[1]*Institute for Space Sciences, Bucharest-Măgurele, R 077125, Romania*

After release from the trap the momentum distribution of an impenetrable gas asymptotically approaches that of a spinless noninteracting Fermi gas in the initial trap. This phenomenon is called dynamical fermionization and, very recently, has been experimentally confirmed in the case of the Lieb-Liniger model in the Tonks-Girardeau regime. We prove analytically and confirm numerically that following the removal of axial confinement the strongly interacting Bose-Fermi mixture exhibits dynamical fermionization and the asymptotical momentum distribution of each component has the same shape as its density profile at $t = 0$. Under a sudden change of the trap frequency to a new non-zero value the dynamics of both fermionic and bosonic momentum distributions presents characteristics which are similar to the case of single component bosons experiencing a similar quench. Our results are derived using a product representation for the correlation functions which, in addition to analytical considerations, can be implemented numerically very easily with complexity which scales polynomially in the number of particles.

## I. INTRODUCTION

The realization that integrable systems do not thermalize is one of the main reasons for the renewed interest in the study of non-equilibrium dynamics of solvable many-body systems [1–4]. A flurry of activity in the last decade has resulted in the introduction of powerful methods like the Quench Action [5, 6] or Generalized Hydrodynamics [7, 8] which were used to theoretically investigate various non-equilibrium scenarios. While a large body of knowledge has accumulated the vast majority of the literature is focused on the case of single component systems [9–28] with very few results reported in the case of multi-component models [29–34]. The need for reliable analytical results on the quench dynamics of integrable systems with internal degrees of freedom cannot be overstated when we take into account that such systems are now routinely realized in laboratories and, in addition, they are expected to present even more intriguing quantum dynamics due to the interplay of the interaction, charge and spin degrees of freedom and statistics of their constituents [35, 36].

The phenomenon of dynamical fermionization (DF) was theoretically predicted in the case of the Tonks-Girardeau (TG) gas (bosonic Lieb-Liniger model with infinite repulsion) in [37] where it was shown that after the release of the trap the momentum distribution evolved from bosonic to fermionic (see also [38–43] and [44] for the anyonic TG gas). These predictions were experimentally verified in a recent experiment [45] which also confirmed the breathing oscillations induced by a sudden change of the trap frequency [38, 46]. While for the TG gas the wavefunctions are obtained via the Bose-Fermi mapping [47, 48] in the case of strongly interacting multi-component systems due to the decoupling of the charge and spin degrees of freedom the wavefunctions have a product form [49–56]. Using such wavefunctions Alam *et al.,* proved DF for the spinor Bose and Fermi gases in [57]. We should point out another distinctive feature of multi-component systems. In the TG (impenetrable) regime the ground state of such systems has a large degeneracy. Averaging over all the degenerate states one obtains the correlators in the spin-incoherent Luttinger liquid (SILL) regime [58–60] which have completely different properties from their counterparts in the Luttinger liquid (LL) phase [61]. Computing the correlators in the LL regime requires the identification of the eigenstate which is continuously connected with the unique ground state of the system at strong but finite interaction. Analytical formulae or determinant representations for the correlators of multi-component systems in the LL regime are exceedingly rare in the literature, one such example will be provided in Sec. III.

In this article we investigate the non-equilibrium dynamics of the harmonically trapped strongly interacting Bose-Fermi mixture (BFM). We report on the derivation of analytical formulae for the correlation functions in the LL regime which we use to investigate, both analytically and numerically, the dynamics after: a) release from the trap and b) sudden change of the trap frequency. The formulae for the one-body reduced density matrices, which constitute one of the main results of this paper, can be implemented numerically very efficiently due to the fact that their numerical complexity depends polynomially in the number of particles and not exponentially like in other approaches. From the analytical point of view we will prove that if we remove the axial confinement the following properties hold: (0) the density profiles at $t = 0$ are proportional to the density of spinless noninteracting fermions (this is the fermionization of the mixture); (1) the asymptotic momentum distribution of each component has the same shape as its density profile at $t = 0$; and (2) the total asymptotic momentum distribution will approach the momentum distribution of spinless noninteracting fermions in the initial trap. Compared with the similar investigation in the case of general bosonic and fermionic spinor gases [57] our explicit expressions for the correlators allow for the derivation of analytical expressions for the initial densities and asymptotic momentum distributions of the individual components (bosonic

and fermionic) and not only for their sums. The numerical investigation of the dynamics after a quench of the trap frequency revealed, rather unexpectedly, that the evolution of the fermionic momentum distribution has characteristics similar to the case of the bosonic TG gas subjected to a similar quench [38] i.e., oscillatory behaviour with additional minima when the gas is maximally compressed [22, 23].

The plan of the article is as follows. In Sec. II we introduce the model and its eigenstates and in Sec. III we present the analytical formulae for the correlation functions. The derivation of the analytical formulae for the correlators, which is based on a new parametrization of the wavefunctions is described in Secs. IV A and IV B. The time evolution of the correlators for the nonequilibrium scenarios considered in this paper is described in Sec. V. In Sec. VI we present the analytical derivation of dynamical fermionization and in Sec. VII we investigate the nonequilibrium dynamics after a sudden change of the trap frequency. We conclude in Sec. VIII. Some examples of the wavefunctions, their orthogonality and normalization and other technical calculations are presented in four Appendices.

## II. MODEL AND EIGENSTATES

We consider a one-dimensional mixture of bosons and spinless fermions with contact interactions in the presence of a time-dependent harmonic potential. The Hamiltonian is

$$\mathcal{H} = \int dx \left[ \sum_{\sigma=\{B,F\}} \frac{\hbar^2}{2m} \partial_x \Psi_\sigma^\dagger \partial_x \Psi_\sigma + V(x,t) \Psi_\sigma^\dagger \Psi_\sigma \right] + \frac{g_{BB}}{2} \Psi_B^\dagger \Psi_B^\dagger \Psi_B \Psi_B + g_{BF} \Psi_B^\dagger \Psi_F^\dagger \Psi_F \Psi_B \,, \tag{1}$$

with $V(x,t) = m\omega^2(t)x^2/2$ the trapping potential, $\omega(t)$ is the time-dependent frequency, $m$ the mass of the particles and $\Psi_B(x)$ and $\Psi_F(x)$ are bosonic and fermionic fields satisfying the commutation relations ($\alpha, \beta = \{B,F\}$), $\Psi_\alpha(x)\Psi_\beta^\dagger(y) = h_{\alpha\beta}\Psi_\beta^\dagger(y)\Psi_\alpha(x) + \delta_{\alpha\beta}\delta(x-y)$, $\Psi_\alpha(x)\Psi_\beta(y) = h_{\alpha\beta}\Psi_\beta(y)\Psi_\alpha(x)$, where $h_{\alpha\beta} = 1$ for $\alpha = \beta = B$; $h_{\alpha\beta} = -1$ for $\alpha = \beta = F$; $h_{\alpha\beta} = -1$ for $\alpha \neq \beta$. We should point out that while we have chosen that the bosonic and fermionic fields anticommute an equally valid choice would have been to choose commutation relations. Each choice results in different wavefunctions but the final results are the same. We will consider the case of impenetrable particles ($g_{BB} = g_{FF} = g = \infty$) which is amenable to an analytical description but we mention that the Hamiltonian (1) is also integrable in the homogeneous case, $V(x,t) = 0$, for any value of $g_{BB} = g_{BF}$ [62, 63].

At $t = 0$ the eigenstates of a system of $N$ particles of which $M$ are bosons are [$\boldsymbol{x} = (x_1, \cdots, x_N)$, $d\boldsymbol{x} = \prod_{i=1}^N dx_i$]

$$|\Phi_{N,M}(\boldsymbol{j}, \boldsymbol{\lambda})\rangle = \int d\boldsymbol{x} \sum_{\alpha_1, \cdots, \alpha_N = \{B,F\}}^{[N,M]} \chi_{N,M}^{\alpha_1 \cdots \alpha_N}(\boldsymbol{x}|\boldsymbol{j}, \boldsymbol{\lambda}) \Psi_{\alpha_N}^\dagger(x_N) \cdots \Psi_{\alpha_1}^\dagger(x_1)|0\rangle \,, \tag{2}$$

where the summation is over all sets of $\alpha$'s of which $M$ are bosonic and $N - M$ are fermionic, constraint which is denoted by $[N,M]$, and $|0\rangle$ is the Fock vacuum satisfying $\Psi_\alpha(x)|0\rangle = \langle 0|\Psi_\alpha^\dagger(x) = 0$ for all $x$ and $\alpha$. In this article $\int dx$ will denote the integral over the real axis. The eigenstates (2) are indexed by two sets of unequal numbers (in each set separately) $\boldsymbol{j} = (j_1, \cdots, j_N)$ and $\boldsymbol{\lambda} = (\lambda_1, \cdots, \lambda_M)$ which describe the charge (orbital) and pseudo-spin degrees of freedom. The normalized wavefunctions are

$$\chi_{N,M}^{\alpha_1 \cdots \alpha_N}(\boldsymbol{x}|\boldsymbol{j}, \boldsymbol{\lambda}) = \frac{1}{N! \, N^{M/2}} \left[ \sum_{P \in S_N} \eta_{N,M}^{\alpha_{P_1} \cdots \alpha_{P_N}}(\boldsymbol{\lambda}) \theta(P\boldsymbol{x}) \right] \det_N [\phi_{j_a}(x_b)]_{a,b=1,\cdots,N} \,, \tag{3}$$

where the sum is over all the permutations $P$ of $N$ elements, $\theta(P\boldsymbol{x}) = \theta(x_{P_1} < \cdots < x_{P_N}) = \prod_{j=2}^N \theta(x_{P_j} - x_{P_{j-1}})$ and $\theta(x)$ the Heaviside function which is 1 when $x \geq 0$ and zero otherwise. The determinant on the right side is defined in terms of the quantum harmonic oscillator wavefunctions

$$\phi_j(x) = \frac{1}{(2^j j!)^{1/2}} \left( \frac{m\omega_0}{\pi\hbar} \right)^{1/4} e^{-\frac{m\omega_0 x^2}{2\hbar}} H_j\left( \sqrt{\frac{m\omega_0}{\hbar}} x \right) \,, \tag{4}$$

with $H_j(x)$ the Hermite polynomials, $\omega_0 = \omega(0)$ and we introduce $l_{HO} = \sqrt{\hbar/(m\omega_0)}$. The pseudo-spin wavefunctions are

$$\eta_{N,M}^{\alpha_1 \cdots \alpha_N}(\boldsymbol{\lambda}) = \det_M \left( e^{i n_a \lambda_b} \right)_{a,b=1,\cdots,M} \,, \tag{5}$$

where $\boldsymbol{\lambda} = (\lambda_1, \cdots, \lambda_M)$ is a subset of cardinality $M$ of the solutions of $e^{i\lambda_a N} = 1$ and $\boldsymbol{n} = (n_1, \cdots, n_M)$ is a set of integers $n_a \in \{1, \cdots, N\}$, describing the positions of the bosons in the ordered set $\{x_1, \cdots, x_N\}$ (for more details and examples see Appendix A).

The wavefunctions (3) represent the generalization of the Bethe ansatz result from [63] in the case of harmonic trapping and it was introduced in a slightly different form in [64]. Let us give some arguments supporting our choice. In first quantization the quantum mechanical Hamiltonian (1) is

$$\mathcal{H} = \sum_{j=1}^{N} \left( -\frac{\hbar^2}{2m} \frac{\partial^2}{\partial x_j^2} + \frac{m\omega_0^2 x_j^2}{2} \right) + g \sum_{i<j} \delta(x_i - x_j), \tag{6}$$

where in the impenetrable case we have $g = \infty$. The wavefunctions (3) are eigenfunctions of the Hamiltonian (6) which by construction vanish when two coordinates coincide (the hard-core condition) and it is easy to see that they are symmetric (antisymmetric) under the exchange of coordinates of two bosons (fermions). Exchanging the coordinates of two fermions affects only the determinant comprised of harmonic oscillator functions and therefore the wavefunctions acquires a minus sign. Exchanging the coordinates of two bosons affects both the charge and pseudo-spin determinants and the two minus signs cancel leaving the wavefunctions unchanged. In addition, the eigenstates (2) are normalized (Appendix B) (we have $\delta_{\boldsymbol{j'j}} = 1$ if the two sets $\boldsymbol{j'}$ and $\boldsymbol{j}$ are equal modulo a permutation) $\langle \Phi_{N',M'}(\boldsymbol{j'}, \boldsymbol{\lambda'}) | \Phi_{N,M}(\boldsymbol{j}, \boldsymbol{\lambda}) \rangle = \delta_{N'N} \delta_{M'M} \delta_{\boldsymbol{j'j}} \delta_{\boldsymbol{\lambda'\lambda}}$ form a complete set and satisfy $\mathcal{H}|\Phi_{N,M}(\boldsymbol{j}, \boldsymbol{\lambda})\rangle = E(\boldsymbol{j})|\Phi_{N,M}(\boldsymbol{j}, \boldsymbol{\lambda})\rangle$ with $E(\boldsymbol{j}) = \sum_{i=1}^{N} \hbar\omega_0(j_i + 1/2)$. We make an important observation. While in (3) we have chosen for the pseudo-spin sector the wavefunctions of free fermions on the lattice (5) in analogy with the Bethe ansatz solution of the homogeneous system [63] we should point out that an equally valid choice would have been any system of functions which ensures that: a) the wavefunctions (3) have the correct symmetry and b) (2) form a complete set of states (for a pedagogical discussion in the case of the Hubbard model in the strong coupling limit which is very similar to our case see Appendix 3.G.1 of [65]).

It is also instructive to look at the wavefunctions (3) in two limiting cases. When the system contains only fermions ($M = 0$) the wavefunctions become

$$\chi_{N,M=0}^{F\cdots F}(\boldsymbol{x}|\boldsymbol{j}) = \frac{1}{N!} \left[ \sum_{P \in S_N} \theta(P\boldsymbol{x}) \right] \det_N [\phi_{j_a}(x_b)], \tag{7}$$

and using $\sum_{P \in S_N} \theta(P\boldsymbol{x}) = 1$ we recognize the Slater determinant for $N$ free fermions in a harmonic trap. In the opposite case when there are no fermions in the system ($M = N$) there is only one pseudo-spin state $\boldsymbol{\lambda} = \frac{2\pi}{N}(0, 1, \cdots, N-1)$ and the wavefunctions are

$$\chi_{N,M=N}^{B\cdots B}(\boldsymbol{x}|\boldsymbol{j}, \boldsymbol{\lambda}) = \frac{\det_N \left( e^{ia\lambda_b} \right)}{N! \, N^{N/2}} \left[ \sum_{P \in S_N} (-1)^P \theta(P\boldsymbol{x}) \right] \det_N [\phi_{j_a}(x_b)]. \tag{8}$$

Now it is easy to see that $\left[ \sum_{P \in S_N} (-1)^P \theta(P\boldsymbol{x}) \right] \det_N [\phi_{j_a}(x_b)] = \left( \prod_{a<b} \text{sign}(x_b - x_a) \right) \det_N [\phi_{j_a}(x_b)]$ which is the wavefunction of $N$ hard-core bosons. The pseudo-spin determinant $\det_N \left( e^{ia\lambda_b} \right)$ (which in this case is of Vandermonde type) is just a constant factor and it can be showed (using the same methods as in Appendix B) that $\det_N \left( e^{-ia\lambda_b} \right) \det_N \left( e^{ia\lambda_b} \right) / N^N = 1$.

At zero temperature the groundstate is highly degenerate and is characterized by $\boldsymbol{j} = 0, 1, \cdots, N-1$ and any $\boldsymbol{\lambda} = (\lambda_1, \cdots, \lambda_M)$ with $\lambda_i$ distinct solutions of $e^{i\lambda_a N} = 1$, $a = 1, \cdots, M$. Averaging over all the degenerate eigenstates would produce results for the SILL regime [58–60]. In this article we will consider the correlators in the LL regime which are computed by considering the state ($M$ is odd) described by

$$\boldsymbol{j} = (0, 1, \cdots, N-1), \tag{9a}$$

$$\boldsymbol{\lambda} = \frac{2\pi}{N} \left( -\frac{M-1}{2} + \frac{N}{2}, \cdots, \frac{N}{2}, \cdots, \frac{M-1}{2} + \frac{N}{2} \right). \tag{9b}$$

The $\boldsymbol{\lambda}$ described by Eq. (9b) is the same as the one identified by Imambekov and Demler for the LL ground state of the homogeneous BFM in [63] and our choice is justified by the fact that the description of the pseudo-spin sector in both the homogeneous and inhomogeneous case is the same as in (3).

## III. ANALYTICAL FORMULAE FOR THE CORRELATION FUNCTIONS

The field-field correlation functions, also known as one-body reduced density matrices, are defined as

$$g_\sigma(\xi_1, \xi_2) = \langle \Phi_{N,M}(\boldsymbol{j}, \boldsymbol{\lambda}) | \Psi_\sigma^\dagger(\xi_1) \Psi_\sigma(\xi_2) | \Phi_{N,M}(\boldsymbol{j}, \boldsymbol{\lambda}) \rangle, \tag{10}$$

with $\sigma \in \{B, F\}$. From the correlation functions we can obtain the real space densities $\rho_\sigma(\xi) \equiv g_\sigma(\xi, \xi)$ and the momentum distributions

$$n_\sigma(p) = \frac{1}{2\pi} \int e^{ip(\xi_1 - \xi_2)/\hbar} g_\sigma(\xi_1, \xi_2)\, d\xi_1 d\xi_2\,, \tag{11}$$

which are important experimental quantities. Introducing $\boldsymbol{\alpha} = (B \cdots B F \cdots F)$, $c_B = (N - M)!(M - 1)!$ and $c_F = (N - M - 1)!M!$ the field-field correlation functions can be written as

$$g_B(\xi_1, \xi_2) = \frac{(N!)^2}{c_B} \int \prod_{j=2}^{N} dx_j\, \bar{\chi}_{N,M}^{\boldsymbol{\alpha}}(\xi_1, x_2, \cdots, x_N | \boldsymbol{j}, \boldsymbol{\lambda}) \chi_{N,M}^{\boldsymbol{\alpha}}(\xi_2, x_2, \cdots, x_N | \boldsymbol{j}, \boldsymbol{\lambda})\,, \tag{12a}$$

$$g_F(\xi_1, \xi_2) = \frac{(N!)^2}{c_F} \int \prod_{j=1}^{N-1} dx_j\, \bar{\chi}_{N,M}^{\boldsymbol{\alpha}}(x_1, \cdots, x_{N-1}, \xi_1 | \boldsymbol{j}, \boldsymbol{\lambda}) \chi_{N,M}^{\boldsymbol{\alpha}}(x_1, \cdots, x_{N-1}, \xi_2 | \boldsymbol{j}, \boldsymbol{\lambda})\,. \tag{12b}$$

Because $g_\sigma(\xi_1, \xi_2) = \overline{g_\sigma(\xi_2, \xi_1)}$ (the bar denotes complex conjugation) it will be sufficient to consider the case $\xi_1 \le \xi_2$. The details of the derivation, which are rather involved, will be presented in Secs. IV A and IV B, here we only present the results. For $\xi_1 \le \xi_2$ and $t = 0$ the correlation functions of the Bose-Fermi mixture can be expressed as sums of products of pseudo-spin and charge functions

$$g_\sigma(\xi_1, \xi_2) = \frac{1}{c_\sigma N^M} \sum_{d_1=1}^{N} \sum_{d_2=d_1}^{N} S_\sigma(d_1, d_2) I(d_1, d_2; \xi_1, \xi_2)\,, \tag{13}$$

where $I(d_1, d_2; \xi_1, \xi_2)$ is the same for both correlators while the pseudo-spin functions $S_{B,F}(d_1, d_2)$ are different. The charge functions have the form $\left(C_N = \prod_{j=0}^{N-1} \left(2^j / \pi^{1/2} j!\right)^{1/2} l_{HO}^{-j-\frac{1}{2}}\right)$

$$I(d_1, d_2; \xi_1, \xi_2) = C_N^2 (-1)^{d_2 - d_1} e^{-\frac{\xi_1^2 + \xi_2^2}{2 l_{HO}^2}} \int_0^{2\pi} \frac{d\psi}{2\pi} e^{-i(d_1 - 1)\psi} \int_0^{2\pi} \frac{d\phi}{2\pi} e^{-i(d_2 - 1)\phi} \det_{N-1} \left[c_{j+k}(\psi, \phi | \xi_1, \xi_2)\right]_{j,k=1,\cdots,N-1}\,, \tag{14}$$

with $c_j(\psi, \phi | \xi_1, \xi_2) = \int t^{j-2} \left[e^{i\psi} f^0(t) + e^{i\phi} f^1(t) + f^2(t)\right] dt$ and $f^{0,1,2}(t) = \mathbf{1}_{0,1,2}\, e^{-t^2/l_{HO}^2}(\xi_1 - t)(\xi_2 - t)$. Here $\mathbf{1}_{0,1,2}$ are the characteristic functions of three intervals $A_0 = (-\infty, \xi_1]$, $A_1 = [\xi_1, \xi_2]$, $A_2 = [\xi_2, +\infty)$ i.e., $\mathbf{1}_i = 1$ when $t \in A_i$ and zero otherwise. The bosonic pseudo-spin functions are

$$S_B(d_1, d_2) = (N - M)!(M - 1)!(-1)^{d_2 - d_1} e^{\frac{2\pi i}{N} \psi_0(d_2 - d_1)} \sum_{r_1=1}^{M} \sum_{r_2=1}^{M-r_1+1} e^{-\frac{2\pi i}{N} \psi_0(r_2 - 1)} f_{r_1-1, r_2-1}^B\,, \tag{15}$$

with $\psi_0 = -(M - 1)/2 + N/2$ and $f_{r_1-1, r_2-1}^B = \int_0^{2\pi} \frac{d\phi}{2\pi} e^{-i(r_1-1)\phi} \int_0^{2\pi} \frac{d\psi}{2\pi} e^{-i(r_2-1)\psi} \det_{M-1} \left[s_B(\phi, \psi, l, j)\right]_{l,j=1,\cdots,M-1}$ where $s_B(\phi, \psi, l, j) = e^{i\phi} s_B^0(d_1, d_2, j, l) + e^{i\psi} s_B^1(d_1, d_2, j, l) + s_B^2(d_1, d_2, j, l)$ and $s_B^{0,1,2}(d_1, d_2, j, l)$ are given by

$$s_B^0(d_1, d_2, j, l) = \sum_{t=1}^{d_1-1} e^{-\frac{2\pi i}{N}(j-l)t} \left(e^{-\frac{2\pi i}{N} t} - e^{-\frac{2\pi i}{N} d_1}\right) \left(e^{\frac{2\pi i}{N} t} - e^{\frac{2\pi i}{N} d_2}\right)\,, \tag{16a}$$

$$s_B^1(d_1, d_2, j, l) = e^{-\frac{2\pi i}{N}(l-1)} \sum_{t=d_1}^{d_2} e^{-\frac{2\pi i}{N}(j-l)t} \left(e^{-\frac{2\pi i}{N} t} - e^{-\frac{2\pi i}{N} d_1}\right) \left(e^{\frac{2\pi i}{N}(t-1)} - e^{\frac{2\pi i}{N} d_2}\right)\,, \tag{16b}$$

$$s_B^2(d_1, d_2, j, l) = \sum_{t=d_2+1}^{N} e^{-\frac{2\pi i}{N}(j-l)t} \left(e^{-\frac{2\pi i}{N} t} - e^{-\frac{2\pi i}{N} d_1}\right) \left(e^{\frac{2\pi i}{N} t} - e^{\frac{2\pi i}{N} d_2}\right)\,. \tag{16c}$$

In the case of the fermionic correlator we have

$$S_F(d_1, d_2) = (N - M - 1)!M!(-1)^{d_2 - d_1} \sum_{r_1=1}^{M+1} \sum_{r_2=1}^{M-r_1+2} e^{-\frac{2\pi i}{N} \psi_0(r_2 - 1)} f_{r_1-1, r_2-1}^F\,, \tag{17}$$

with $f^F_{r_1-1,r_2-1} = \int_0^{2\pi} \frac{d\phi}{2\pi} e^{-i(r_1-1)\phi} \int_0^{2\pi} \frac{d\psi}{2\pi} e^{-i(r_2-1)\psi} \det_M [s_F(\phi,\psi,l,j)]_{l,j=1,\cdots,M}$ where $s_F(\phi,\psi,l,j) = e^{i\phi} s_F^0(d_1,d_2,j,l) + e^{i\psi} s_F^1(d_1,d_2,j,l) + s_F^2(d_1,d_2,j,l)$ with $s_F^{0,1,2}(d_1,d_2,j,l)$ simple functions

$$s_F^0(d_1,d_2,j,l) = \sum_{t=1}^{d_1-1} e^{-\frac{2\pi i}{N}(j-l)t} , \tag{18a}$$

$$s_F^1(d_1,d_2,j,l) = e^{-\frac{2\pi i}{N}(l-1)} \sum_{t=d_1+1}^{d_2} e^{-\frac{2\pi i}{N}(j-l)t} , \tag{18b}$$

$$s_F^2(d_1,d_2,j,l) = \sum_{t=d_2+1}^{N} e^{-\frac{2\pi i}{N}(j-l)t} . \tag{18c}$$

Even though Eqs. (13), (14), (15) and (17) may seem cumbersome they are in fact extremely efficient from the numerical point of view due to the fact that their complexities scale polynomially and not exponentially in the number of particles. In addition, the pseudo-spin functions (15) and (17) do not depend on the space separation so they need to be computed only once for given $N$ and $M$. The functions $c_j(\psi,\phi|\xi_1,\xi_2)$ appearing in the definition of the charge function Eq. (14) can be expressed in terms of the complete and incomplete Gamma functions and in fact we need only $2N-3$ evaluation of these integrals for a given space separation because the determinant appearing in the definition is of Hankel type of dimension $N-1$ (the numerical evaluation of a determinant of dimension $N$ takes $N^3$ operations). Using the fractional Fast Fourier Transform for the calculation of the double integrals [66, 67] the computation of the correlation functions for 30 particles and given $\xi_1$ and $\xi_2$ takes less than 4 seconds using an interpreted language on a common laptop. For other methods of numerically computing the correlators in multi-component systems see [32, 68–73, 75, 76, 91].

## IV.  NEW PARAMETRIZATION FOR THE WAVEFUNCTIONS AND DERIVATION OF THE ANALYTICAL FORMULAE FOR THE CORRELATORS

The method used to derive the product formulae for the correlation functions presented in Sec. III can be understood as the generalization to the inhomogeneous case of the technique used by Imambekov and Demler [63] in their study of the homogeneous BFM. The main ingredient is a new parametrization of the wavefunctions with $\boldsymbol{\alpha} = (B\cdots BF\cdots F)$, called the canonical ordering, which appears in the expressions (12) for the correlation functions. We introduce a set of $N$ ordered variables

$$Z = \{-\infty \leq z_1 \leq z_2 \leq \cdots \leq z_N \leq +\infty\}, \tag{19}$$

which describes the positions of the particles independent of their statistics. For a given set of $\boldsymbol{x} = (x_1,\cdots,x_N)$ exchanging the position of any pair of particles means they will be described by the same ordered set $\boldsymbol{z} = (z_1,\cdots,z_N)$. Because originally the wavefunction was defined in $\mathbb{R}^N$ and the $z$ variables belong to (19) we need to introduce an additional set of $N$ variables denoted by $\boldsymbol{y} = (y_1,\cdots,y_N)$ which specify the positions of the $i$-th particle in the ordered set $z_1 \leq \cdots \leq z_N$. $y_1,\cdots,y_M$ specify the positions of the bosons and $y_{M+1},\cdots,y_N$ the position of the fermions and they satisfy $z_{y_i} = x_i$. For example, if we consider the $(5,2)$-sector ($N=5$ and $M=2$) with $x_3 < x_1 < x_2 < x_4 < x_5$, then $\boldsymbol{z} \equiv (z_1,z_2,z_3,z_4,z_5) = (x_3,x_1,x_2,x_4,x_5)$ and $\boldsymbol{y} = (2,3,1,4,5)$. Note that the $y_1,\cdots,y_M$ variables are equivalent with the $n$'s in the original definition (Sec. II) for the canonical ordering. In the new parametrization the wavefunction takes the form

$$\chi_{N,M}(\boldsymbol{z},\boldsymbol{y}|\boldsymbol{j},\boldsymbol{\lambda}) = \frac{1}{N!N^{M/2}}(-1)^{\boldsymbol{y}} \det_M \left(e^{iy_a\lambda_b}\right) \det_N [\phi_{j_a}(z_b)] , \tag{20}$$

with $(-1)^{\boldsymbol{y}} = \prod_{N \geq i > k \geq 1} \text{sign}(y_i - y_k)$. It is important to note that the first determinant depends only on the positions of the bosons $y_1,\cdots,y_M$, the dependence on $y_{M+1},\cdots,y_N$ appears only in the $(-1)^{\boldsymbol{y}}$ factor. This parametrization has the advantage of making clear the factorization of the pseudo-spin and charge degrees of freedom in the expressions for the correlators (12) as it is shown in Appendix C.

## A. The Bose-Bose correlator

The results of Appendix C show that in the new parametrization the Bose-Bose correlator in the region $\xi_1 \leq \xi_2$ can be written in a factorized form as

$$g_B(\xi_1, \xi_2) = \frac{1}{(N-M)!(M-1)!N^M} \sum_{d_1=1}^{N} \sum_{d_2=d_1}^{N} S_B(d_1, d_2) I(d_1, d_2; \xi_1, \xi_2), \tag{21}$$

with

$$I(d_1, d_2; \xi_1, \xi_2) = \int_{Z_{d_1,d_2}(\xi_1,\xi_2)} \prod_{j=1, j\neq d_1}^{N} dz_j \det_N \left[ \bar{\phi}_{j_a}(z_b) \right] \det_N \left[ \phi_{j_a}(z_b') \right], \tag{22}$$

and

$$S_B(d_1, d_2) = \sum_{\boldsymbol{y} \in Y(y_1=d_1)} (-1)^{\boldsymbol{y}+\boldsymbol{y}'} \det_M \left( e^{-iy_a \lambda_b} \right) \det_M \left( e^{iy_a' \lambda_b} \right). \tag{23}$$

In (22) $Z_{d_1,d_2}(\xi_1, \xi_2)$ is defined as

$$Z_{d_1,d_2}(\xi_1, \xi_2) = \left\{ -\infty \leq z_1 \leq \cdots \leq z_{d_1-1} \leq \xi_1 \leq z_{d_1+1} \leq \cdots \leq z_{d_2} \leq \xi_2 \leq z_{d_2+1} \leq \cdots \leq z_N \leq +\infty \right\}, \tag{24}$$

and $\boldsymbol{z}'$ satisfies the constraint

$$-\infty \leq z_1' = z_1 \leq \cdots \leq z_{d_1-1}' = z_{d_1-1} \leq \xi_1 \leq z_{d_1}' = z_{d_1+1} \leq \cdots$$
$$\cdots \leq z_{d_2-1}' = z_{d_2} \leq \xi_2 \leq z_{d_2+1}' = z_{d_2+1} \leq \cdots \leq z_N' = z_N \leq +\infty, \tag{25}$$

while in (23) $Y(y_1 = d) = \{ \boldsymbol{y} \in S_N | \text{ with } y_1 = d \}$, and the connection between $\boldsymbol{y}$ and $\boldsymbol{y}'$ is given by

$$\begin{aligned}
y_1' &= d_2, \quad y_1 = d_1, \\
y_i' &= y_i \quad \text{for } y_i < d_1, \\
y_i' &= y_i - 1 \quad \text{for } d_1 < y_i \leq d_2, \\
y_i' &= y_i, \quad \text{for } d_2 < y_i.
\end{aligned} \tag{26}$$

In the region $\xi_2 < \xi_1$ the correlator can be obtained from the previous expression (21) through complex conjugation $g_B(\xi_1, \xi_2) = \overline{g_B(\xi_2, \xi_1)}$. While (21), (22) and (23) describe the Bose-Bose correlator in any eigenstate of the system below we will be interested in computing $g_B(\xi_1, \xi_2)$ in the LL groundstate which is characterized by (9). We will start with the computation of the charge functions.

### 1. Calculation of the charge functions $I(d_1, d_2; \xi_1, \xi_2)$

In the LL groundstate $\boldsymbol{j} = (0, 1, \cdots, N-1)$. This means that the determinants appearing in (22) are of Vandermonde type and we can apply Vandermonde's formula

$$\det_N \left[ p_{a-1}(z_b) \right]_{a,b=1,\cdots,N} = \prod_{1 \leq a < b \leq N} (z_b - z_i), \tag{27}$$

valid for $p_a(z)$ monic polynomials of degree $a$ (a polynomial in $z$ of degree $p$ is monic if the coefficient of $z^p$ is 1). The harmonic oscillator wavefunctions can be written as

$$\phi_j(z) = c_j e^{-\frac{z^2}{2l_{HO}^2}} \tilde{H}_j(z), \quad c_j = \left( \frac{2^j}{\pi^{1/2} j!} \right)^{1/2} \frac{1}{l_{HO}^{j+\frac{1}{2}}}, \tag{28}$$

with $\tilde{H}_j(z) = l_{HO}^j H_j(z/l_{HO})/2^j$ monic polynomials. Using Vandermonde's formula (27) we find

$$\det_N \left[ \phi_{j_a}(z_b) \right] = \left( \prod_{j=0}^{N-1} c_j \right) e^{-\frac{1}{2l_{HO}^2}(z_1^2 + \cdots + z_N^2)} \prod_{1 \leq a < b \leq N} (z_b - z_a), \tag{29}$$

with $z_{d_1} = \xi_1$. This expression can be rewritten in a more computationally friendly way by moving the $d_1$ column (containing $z_{d_1} = \xi_1$) to the first position (this produces a $(-1)^{d_1-1}$ factor) and introducing $N-1$ new variables of integration $t_i = z_i$ for $i < d_1$ and $t_i = z_{i+1}$ for $i > d_1$. We obtain

$$
\det_N \left[ \bar{\phi}_{j_a}(z_b) \right] = C_N(-1)^{d_1-1} e^{-\frac{1}{2l_{HO}^2}(\xi_1^2 + t_1^2 + \cdots + t_{N-1}^2)} \prod_{i=1}^{N-1} (t_i - \xi_1) \prod_{1 \le a < b \le N-1} (t_b - t_a),
$$

$$
= C_N(-1)^{d_1-1} e^{-\frac{1}{2l_{HO}^2}(\xi_1^2 + t_1^2 + \cdots + t_{N-1}^2)} \prod_{i=1}^{N-1} (t_i - \xi_1) \det_{N-1} \left( t_a^{b-1} \right)_{a,b=1,\cdots,N-1}, \tag{30}
$$

where we have introduced $C_N = \prod_{j=0}^{N-1} c_j$. In a similar fashion we have

$$
\det_N \left[ \phi_{j_a}(z_b') \right] = C_N(-1)^{d_2-1} e^{-\frac{1}{2l_{HO}^2}(\xi_2^2 + t_1^2 + \cdots + t_{N-1}^2)} \prod_{i=1}^{N-1} (t_i - \xi_2) \det_{N-1} \left( t_a^{b-1} \right)_{a,b=1,\cdots,N-1}. \tag{31}
$$

In the new variables the integration subspace $Z_{d_1,d_2}(\xi_1, \xi_2)$ is

$$
\{ -\infty \le \underbrace{t_1 \le \cdots \le t_{d_1-1}}_{d_1-1} \le \xi_1 \le \underbrace{t_{d_1} \le \cdots \le t_{d_2-1}}_{d_2-d_1} \le \xi_2 \le \underbrace{t_{d_2} \le \cdots \le t_{N-1}}_{N-d_2} \le +\infty \}. \tag{32}
$$

Remembering that $\boldsymbol{z'}$ and $\boldsymbol{z}$ satisfy the constraint (25) (this also applies to $\boldsymbol{t'}$ and $\boldsymbol{t}$ after changing the indices) we see that the product of (30) and (31) is invariant by exchanging $t_i$ and $t_j$ with $-\infty \le t_i, t_j \le \xi_1$ and is zero when $t_i = t_j$ in the same region. The same property holds for the regions $\xi_1 \le t_i, t_j \le \xi_2$ and $\xi_2 \le t_i, t_j \le +\infty$ which means that the integral over $Z_{d_1,d_2}(\xi_1, \xi_2)$ can be written as an integral over

$$
T_{d_1,d_2}(\xi_1, \xi_2) = \{ -\infty \le t_1, \cdots, t_{d_1-1} \le \xi_1 \le t_{d_1}, \cdots, t_{d_2-1} \le \xi_2 \le t_{d_2}, \cdots, t_{N-1} \le +\infty \}, \tag{33}
$$

multiplied by $1/[(d_1-1)!(d_2-d_1)!(N-d_2)!]$. Expanding the determinants in (30) and (31) we obtain

$$
I(d_1, d_2; \xi_1, \xi_2) = \frac{C_N^2(-1)^{d_2-d_1} e^{-\frac{1}{2l_{HO}^2}(\xi_1^2 + \xi_2^2)}}{(d_1-1)!(d_2-d_1)!(N-d_2)!} \int_{T_{d_1,d_2}(\xi_1,\xi_2)} \prod_{j=1}^{N-1} dt_j \sum_{P \in S_{N-1}} \sum_{P' \in S_{N-1}} (-1)^{P+P'}
$$

$$
\times \prod_{i=1}^{N-1} \left( t_i^{P_i-1} t_i^{P'_i-1} e^{-t_i^2/l_{HO}^2} (t_i - \xi_1)(t_i - \xi_2) \right). \tag{34}
$$

Introducing three functions

$$
\begin{aligned}
f^0(\xi_1, \xi_2, t) &= e^{-t^2/l_{HO}^2}(\xi_1 - t)(\xi_2 - t) &&\text{for } -\infty < t < \xi_1, &&& 0 \text{ otherwise}, \\
f^1(\xi_1, \xi_2, t) &= e^{-t^2/l_{HO}^2}(\xi_1 - t)(\xi_2 - t) &&\text{for } \xi_1 < t < \xi_2, &&& 0 \text{ otherwise}, \\
f^2(\xi_1, \xi_2, t) &= e^{-t^2/l_{HO}^2}(\xi_1 - t)(\xi_2 - t) &&\text{for } \xi_2 < t < +\infty, &&& 0 \text{ otherwise},
\end{aligned} \tag{35}
$$

and writing $P' = QP$, which can be done for any permutations $P$ and $P'$, we find

$$
I(d_1, d_2; \xi_1, \xi_2) = \frac{C_N^2(-1)^{d_2-d_1} e^{-\frac{1}{2l_{HO}^2}(\xi_1^2 + \xi_2^2)}}{(d_1-1)!(d_2-d_1)!(N-d_2)!} \sum_{P \in S_{N-1}} \sum_{Q \in S_{N-1}} (-1)^Q \left( \prod_{i=1}^{d_1-1} \int t_i^{P_i+QP_i-2} f^0(\xi_1, \xi_2, t_i) \, dt_i \right)
$$

$$
\times \left( \prod_{i=d_1}^{d_2-1} \int t_i^{P_i+QP_i-2} f^1(\xi_1, \xi_2, t_i) \, dt_i \right) \left( \prod_{i=d_2}^{N-1} \int t_i^{P_i+QP_i-2} f^2(\xi_1, \xi_2, t_i) \, dt_i \right). \tag{36}
$$

Using two 'phase' variables $\psi$ and $\phi$ (this is a similar trick as the one employed in [50] and [63]) this result can be written as

$$
I(d_1, d_2; \xi_1, \xi_2) = C_N^2(-1)^{d_2-d_1} e^{-\frac{1}{2l_{HO}^2}(\xi_1^2 + \xi_2^2)} \int_0^{2\pi} \frac{d\psi}{2\pi} e^{-i(d_1-1)\psi} \int_0^{2\pi} \frac{d\phi}{2\pi} e^{-i(d_2-1)\phi}
$$

$$\times \left( \sum_{Q \in S_{N-1}} (-1)^Q \prod_{i=1}^{N-1} \int v_i^{i+Q_i-2} \left( e^{i\psi} f^0(\xi_1, \xi_2, v_i) + e^{i\phi} f^1(\xi_1, \xi_2, v_i) + f^2(\xi_1, \xi_2, v_i) \right) dv_i \right). \quad (37)$$

The equivalence of (36) and (37) can be seen as follows. In (36) the sum over $P \in S_{N-1}$ can be decomposed as a sum over $C_{d_1-1}^{N-1} C_{d_2-d_1}^{N-1-(d_1-1)}$ terms which give the same result with 'degeneracy' given by $(d_1 - 1)!(d_2 - d_1)!((N - d_2)!$. This is because if we choose a permutation with the first $d_1 - 1$ elements given by $P_1, \cdots, P_{d_1-1}$ all the permutations of this set leaves the first parenthesis in (36). The second parenthesis is invariant by permuting the next $d_2 - d_1$ elements $P_{d_1}, \cdots, P_{d_2-1}$ and the last parenthesis will also give $(N - d_2)!$ equal terms. The integral in (37) over the product will also give $C_{d_1-1}^{N-1} C_{d_2-d1}^{N-1-(d_1-1)}$ terms which are obtaining by selecting $d_1 - 1$ terms containing $e^{i\psi}$ out of the $N - 1$ $v_i$'s and then selecting $d_2 - 1$ terms containing $e^{i\phi}$ out of the remaining $N - 1 - (d_1 - 1)$ $v_i$'s. They are in one-to-one connection with the similar terms from (36). Eq. (37) can also be written as

$$I(d_1, d_2; \xi_1, \xi_2) = C_N^2 (-1)^{d_2-d_1} e^{-\frac{1}{2l_{HO}^2}(\xi_1^2 + \xi_2^2)} \int_0^{2\pi} \frac{d\psi}{2\pi} e^{-i(d_1-1)\psi} \int_0^{2\pi} \frac{d\phi}{2\pi} e^{-i(d_2-1)\phi}$$

$$\times \det_{N-1} \begin{vmatrix} c_0(\psi, \phi) & c_1(\psi, \phi) & c_2(\psi, \phi) & \cdots & c_{N-2}(\psi, \phi) \\ c_1(\psi, \phi) & c_2(\psi, \phi) & c_3(\psi, \phi) & \cdots & c_{N-1}(\psi, \phi) \\ \vdots & \vdots & \vdots & \ddots & \vdots \\ c_{N-2}(\psi, \phi) & c_{N-1}(\psi, \phi) & c_N(\psi, \phi) & \cdots & c_{2N-4}(\psi, \phi) \end{vmatrix}, \quad (38)$$

with

$$c_j(\psi, \phi | \xi_1, \xi_2) = \int t^{j-2} \left( e^{i\psi} f^0(\xi_1, \xi_2, t) + e^{i\phi} f^1(\xi_1, \xi_2, t) + f^2(\xi_1, \xi_2, t) \right) dt, \quad (39)$$

which is Eq. (14) of Sec. III.

### 2. Calculation of the pseudo-spin functions $S_B(d_1, d_2)$

Using the relation (26) between $\boldsymbol{y}$ and $\boldsymbol{y}'$ and taking into account that $(-1)^{\boldsymbol{y}}$ measures the number of transpositions necessary to order the set $\boldsymbol{y}$ it can be shown that for any $\boldsymbol{y} \in S(y_1 = d_1)$ we have $(-1)^{\boldsymbol{y}+\boldsymbol{y}'} = (-1)^{d_2-d_1}$. The pseudo-spin function $S_B(d_1, d_2)$ depends on $y_{M+1}, \cdots, y_N$ only through this sign prefactor so we can write

$$S_B(d_1, d_2) = (N - M)! \sum_{y_2=1}^N \cdots \sum_{y_M=1}^N \det_M \left( e^{-iy_a \lambda_b} \right) \det_M \left( e^{iy_a' \lambda_b} \right). \quad (40)$$

where we have used the fact that the determinants are zero when two of the $y$'s are equal. In the LL groundstate (9) the $\lambda$'s are are also equidistant so we can again write the determinants in a Vandermonde form. We obtain $(\psi_0 = -(M+1)/2 + N/2)$

$$\det_M \left( e^{-iy_a \lambda_b} \right) = e^{-\frac{2\pi i}{N} \psi_0 (d_1 + y_2 + \cdots + y_M)} \det_M \left( e^{-\frac{2\pi i}{N}(l-1)y_j} \right)_{l,j=1,\cdots,M},$$

$$= e^{-\frac{2\pi i}{N} \psi_0 (d_1 + y_2 + \cdots + y_M)} \prod_{1 \leq j_1 < j_2 \leq M} \left( e^{-\frac{2\pi i}{N} y_{j_2}} - e^{-\frac{2\pi i}{N} y_{j_1}} \right), \quad (41)$$

and

$$\det_M \left( e^{iy_a' \lambda_b} \right) = e^{\frac{2\pi i}{N} \psi_0 (d_2 + y_2' + \cdots + y_M')} \det_M \left( e^{\frac{2\pi i}{N}(l-1)y_j'} \right)_{l,j=1,\cdots,M},$$

$$= e^{\frac{2\pi i}{N} \psi_0 (d_2 + y_2' + \cdots + y_M')} \prod_{1 \leq j_1 < j_2 \leq M} \left( e^{\frac{2\pi i}{N} y_{j_2}'} - e^{\frac{2\pi i}{N} y_{j_1}'} \right). \quad (42)$$

Like in the previous section we want to extract a determinant of dimension $(M - 1) \times (M - 1)$ multiplied with the contributions of $y_1$ and $y_1'$. Introducing the new variables $t_i = y_{i+1}$ and $t_i' = y_{i+1}'$ for $i \in \{1, \cdots, M - 1\}$ we find

$$\det_M \left( e^{-iy_a \lambda_b} \right) = e^{-\frac{2\pi i}{N} \psi_0 (d_1 + t_1 + \cdots + t_{M-1})} \prod_{i=1}^{M-1} \left( e^{-\frac{2\pi i}{N} t_i} - e^{-\frac{2\pi i}{N} d_1} \right) \det_{M-1} \left( e^{-\frac{2\pi i}{N}(l-1)t_j} \right)_{l,j=1,\cdots,M-1}, \quad (43)$$

$$\det_M \left( e^{i y'_a \lambda_b} \right) = e^{\frac{2\pi i}{N} \psi_0 (d_2 + t'_1 + \cdots + t'_{M-1})} \prod_{i=1}^{M-1} \left( e^{\frac{2\pi i}{N} t'_i} - e^{\frac{2\pi i}{N} d_2} \right) \det_{M-1} \left( e^{\frac{2\pi i}{N} (l-1) t'_j} \right)_{l,j=1,\cdots,M-1}. \tag{44}$$

We break the summation appearing in (40) as follows: we choose $r_1 - 1$ of the $t_i$ satisfying $t_i < d_1$, $r_2 - 1$ of the $t_i$ satisfying $d_1 \le t_i \le d_2$ and the rest of the $(M-1) - (r_1 - 1) - (r_2 - 1)$ variables will satisfy $d_2 < t_i$. Then, we have

$$S_B(d_1, d_2) = (N - M)! \sum_{r_1=1}^{M} \sum_{r_2=1}^{M-(r_1-1)} C_{r_1-1}^{M-1} C_{r_2-1}^{(M-1)-(r_1-1)} S_B(d_1, d_2; r_1, r_2), \tag{45}$$

with the domain of summation for $S_B(d_1, d_2; r_1, r_2)$ being

$$T(d_1, d_2; r_1, r_2) = \{ 1 \le t_1, \cdots, t_{r_1-1} < d_1 \le t_{r_1}, \cdots, t_{r_1+r_2-2} \le d_2 < t_{r_1+r_2-1}, \cdots, t_{M-1} \le N \}. \tag{46}$$

Expanding the determinants in (43) and (44) we find (note that for some values of $r_1$ and $r_2$ the $S_B(d_1, d_2; r_1, r_2)$ functions will be zero)

$$S_B(d_1, d_2; r_1, r_2) = (-1)^{d_2 - d_1} e^{-\frac{2\pi i}{N} \psi_0 (d_1 - d_2 + r_2 - 1)} \sum_{P' \in S_{M-1}} \sum_{P \in S_{M-1}} (-1)^{P' + P}$$

$$\times \prod_{i=1}^{r_1-1} \left[ \sum_{t_i=1}^{d_1-1} e^{-\frac{2\pi i}{N} [(P'_i-1)t_i - (P_i-1)t_i]} \left( e^{-\frac{2\pi i}{N} t_i} - e^{-\frac{2\pi i}{N} d_1} \right) \left( e^{\frac{2\pi i}{N} t_i} - e^{\frac{2\pi i}{N} d_2} \right) \right]$$

$$\times \prod_{i=r_1}^{r_1+r_2-2} \left[ \sum_{t_i=d_1}^{d_2} e^{-\frac{2\pi i}{N} [(P'_i-1)t_i - (P_i-1)(t_i-1)]} \left( e^{-\frac{2\pi i}{N} t_i} - e^{-\frac{2\pi i}{N} d_1} \right) \left( e^{\frac{2\pi i}{N} (t_i-1)} - e^{\frac{2\pi i}{N} d_2} \right) \right]$$

$$\times \prod_{i=r_1+r_2-1}^{M-1} \left[ \sum_{t_i=d_2+1}^{N} e^{-\frac{2\pi i}{N} [(P'_i-1)t_i - (P_i-1)t_i]} \left( e^{-\frac{2\pi i}{N} t_i} - e^{-\frac{2\pi i}{N} d_1} \right) \left( e^{\frac{2\pi i}{N} t_i} - e^{\frac{2\pi i}{N} d_2} \right) \right]. \tag{47}$$

Using the functions $s_B^{0,1,2}(d_1, d_2, j, l)$ defined in (16) and writing $P' = QP$ then (47) can be written as

$$S_B(d_1, d_2; r_1, r_2) = (-1)^{d_2 - d_1} e^{-\frac{2\pi i}{N} \psi_0 (d_1 - d_2 + r_2 - 1)} \sum_{P \in S_{M-1}} \sum_{Q \in S_{M-1}} (-1)^Q \prod_{i=1}^{r_1-1} s_B^0(d_1, d_2, QP_i, P_i)$$

$$\times \prod_{i=r_1}^{r_1+r_2-2} s_B^1(d_1, d_2, QP_i, P_i) \prod_{i=r_1+r_2-1}^{M-1} s_B^2(d_1, d_2, QP_i, P_i), \tag{48}$$

and

$$S_B(d_1, d_2) = (N - M)! \sum_{r_1=1}^{M} \sum_{r_2=1}^{M-(r_1-1)} \frac{(M-1)!}{(M - r_1 - r_2 + 1)!(r_1-1)!(r_2-1)!} S_B(d_1, d_2; r_1, r_2). \tag{49}$$

Like in the previous section the spin function can be expressed as a double integral over a determinant of dimension $(M-1) \times (M-1)$ by introducing two 'phases' with the result

$$S_B(d_1, d_2) = (N - M)!(M-1)!(-1)^{d_2 - d_1} \sum_{r_1=1}^{M} \sum_{r_2=1}^{M-(r_1-1)} e^{-\frac{2\pi i}{N} \psi_0 (d_1 - d_2 + r_2 - 1)} \int_0^{2\pi} \frac{d\phi}{2\pi} e^{-(r_1-1)\phi} \int_0^{2\pi} \frac{d\psi}{2\pi} e^{-(r_2-1)\psi}$$

$$\times \det_{M-1} \begin{vmatrix} s_B(\phi, \psi, 1, 1) & s_B(\phi, \psi, 2, 1) & \cdots & s_B(\phi, \psi, M-1, 1) \\ s_B(\phi, \psi, 1, 2) & s_B(\phi, \psi, 2, 2) & \cdots & s_B(\phi, \psi, M-1, 2) \\ \vdots & \vdots & \ddots & \vdots \\ s_B(\phi, \psi, 1, M-1) & s_B(\phi, \psi, 2, M-1) & \cdots & s_B(\phi, \psi, M-1, M-1) \end{vmatrix}, \tag{50}$$

where

$$s_B(\phi, \psi, l, j) = e^{i\phi} s_B^0(d_1, d_2, j, l) + e^{i\psi} s_B^1(d_1, d_2, j, l) + s_B^2(d_1, d_2, j, l). \tag{51}$$

Because the determinant appearing in (50) is of dimension $M - 1$ we have $\det[\phi, \psi] = \sum_{m=0}^{M-1} \sum_{n=0}^{M-1} f_{m,n} e^{im\phi} e^{in\psi}$. Using this expression in (50) we obtain our final expression for the bosonic pseudo-spin functions (Eq. (15) of Sec. III)

$$S_B(d_1, d_2) = (N - M)!(M-1)!(-1)^{d_2 - d_1} e^{\frac{2\pi i}{N} \psi_0 (d_2 - d_1)} \sum_{r_1=1}^{M} \sum_{r_2=1}^{M-(r_1-1)} e^{-\frac{2\pi i}{N} \psi_0 (r_2 - 1)} f_{r_1-1, r_2-1}. \tag{52}$$

## B. The Fermi-Fermi correlator

From Appendix C the Fermi-Fermi correlator can be written in a factorized form as

$$g_F(\xi_1, \xi_2) = \frac{1}{(N-M-1)!M!N^M} \sum_{d_1=1}^{N} \sum_{d_2=d_1}^{N} S_F(d_1, d_2) I(d_1, d_2; \xi_1, \xi_2) \,, \tag{53}$$

with $I(d_1, d_2; \xi_1, \xi_2)$ given by the same expression as in the case of the Bose-Bose correlators (22) and the pseudo-spin function defined as

$$S_F(d_1, d_2) = \sum_{\boldsymbol{y} \in Y(y_N = d_1)} (-1)^{\boldsymbol{y} + \boldsymbol{y}'} \det_M \left( e^{-iy_a \lambda_b} \right) \det_M \left( e^{iy'_a \lambda_b} \right) \,, \tag{54}$$

with the connection between $\boldsymbol{y}$ and $\boldsymbol{y}'$ given by

$$\begin{aligned}
y'_N &= d_2 \,, \quad y_N = d_1 \,, \\
y'_i &= y_i \quad \text{for } y_i < d_1 \,, \\
y'_i &= y_i - 1 \quad \text{for } d_1 < y_i \leq d_2 \,, \\
y'_i &= y_i \,, \quad \text{for } d_2 < y_i \,.
\end{aligned} \tag{55}$$

Note that in this case $y_1, \cdots, y_M$ and $y'_1, \cdots, y'_M$ cannot take the values $d_1$ and $d_2$.

### 1. Calculation of the pseudo-spin functions $S_F(d_1, d_2)$

The $I(d_1, d_2; \xi_1, \xi_2)$ function has been calculated in Sec. IV A 1 so we need to focus only on $S_F(d_1, d_2)$. Like in the bosonic case we have $(-1)^{\boldsymbol{y}+\boldsymbol{y}'} = (-1)^{d_2 - d_1}$ and the spin function depends on $y_{M+1}, \cdots, y_{N-1}$ only through this sign prefactor so we can write

$$S_F(d_1, d_2) = (N-M-1)! \sum_{y_1=1, y_1 \neq d_1}^{N} \cdots \sum_{y_M=1, y_M \neq d_1}^{N} \det_M \left( e^{-iy_a \lambda_b} \right) \det_M \left( e^{iy'_a \lambda_b} \right) \,. \tag{56}$$

Introducing new variables $t_i = y_i$ and $t'_i = y'_i$ for $i \in \{1, \cdots, M\}$ (this is superfluous but it is done in order to use the same notations as in the bosonic case) the determinants can be written in Vandermonde form

$$\det_M \left( e^{-iy_a \lambda_b} \right) = e^{-\frac{2\pi i}{N} \psi_0 (t_1 + \cdots + t_M)} \det_M \left( e^{-\frac{2\pi i}{N}(l-1)t_j} \right)_{l,j=1,\cdots,M} \,, \tag{57}$$

$$\det_M \left( e^{iy'_a \lambda_b} \right) = e^{\frac{2\pi i}{N} \psi_0 (t'_1 + \cdots + t'_M)} \det_M \left( e^{\frac{2\pi i}{N}(l-1)t'_j} \right)_{l,j=1,\cdots,M} \,. \tag{58}$$

The summation appearing in (56) is broken as follows: we choose $r_1 - 1$ of the $t_i$'s who will satisfy $t_i < d_1$, $r_2 - 1$ of the variables will satisfy $d_1 < t_i \leq d_2$ and the rest of the $M - (r_1 - 1) - (r_2 - 1)$ variables will satisfy $d_2 < t_i$. The fermionic pseudo-spin function then can be written as

$$S_F(d_1, d_2) = (N-M-1)! \sum_{r_1=1}^{M+1} \sum_{r_2=1}^{M+1-(r_1-1)} C_{r_1-1}^{M} C_{r_2-1}^{M-(r_1-1)} S_F(d_1, d_2; r_1, r_2) \,, \tag{59}$$

with the domain of summation for $S_F(d_1, d_2; r_1, r_2)$ being

$$T(d_1, d_2; r_1, r_2) = \{1 \leq t_1, \cdots, t_{r_1-1} < d_1 < t_{r_1}, \cdots, t_{r_1+r_2-2} \leq d_2 < t_{r_1+r_2-1}, \cdots, t_M \leq N\} \,. \tag{60}$$

Using (57) and (58) we find

$$S_B(d_1, d_2; r_1, r_2) = (-1)^{d_2-d_1} e^{-\frac{2\pi i}{N} \psi_0 (r_2-1)} \sum_{P' \in S_M} \sum_{P \in S_M} (-1)^{P'+P} \prod_{i=1}^{r_1-1} \left( \sum_{t_i=1}^{d_1-1} e^{-\frac{2\pi i}{N}[(P'_i-1)t_i - (P_i-1)t_i]} \right)$$

$$\times \prod_{i=r_1}^{r_1+r_2-2} \left( \sum_{t_i=d_1+1}^{d_2} e^{-\frac{2\pi i}{N}[(P'_i-1)t_i-(P_i-1)(t_i-1)]} \right) \prod_{i=r_1+r_2-1}^{M} \left( \sum_{t_i=d_2+1}^{N} e^{-\frac{2\pi i}{N}[(P'_i-1)t_i-(P_i-1)t_i]} \right). \quad (61)$$

Writing $P' = QP$ then (61) can be written as

$$S_F(d_1,d_2;r_1,r_2) = (-1)^{d_2-d_1} e^{-\frac{2\pi i}{N}\psi_0(r_2-1)} \sum_{P\in S_M} \sum_{Q\in S_M} (-1)^Q \prod_{i=1}^{r_1-1} s_F^0(d_1,d_2,QP_i,P_i)$$

$$\times \prod_{i=r_1}^{r_1+r_2-2} s_F^1(d_1,d_2,QP_i,P_i) \prod_{i=r_1+r_2-1}^{M} s_F^2(d_1,d_2,QP_i,P_i), \quad (62)$$

with $s_F^0(d_1,d_2,j,l)$ defined in (18) and

$$S_F(d_1,d_2) = (N-M-1)! \sum_{r_1=1}^{M+1} \sum_{r_2=1}^{M-r_1+2} \frac{M!}{(M-r_1-r_2+2)!(r_1-1)!(r_2-1)!} S_F(d_1,d_2;r_1,r_2). \quad (63)$$

Introduction of two 'phases' allows to express the spin function as a double integral over a determinant of dimension $M \times M$ (compare with the bosonic case)

$$S_F(d_1,d_2) = (N-M-1)!M!(-1)^{d_2-d_1} \sum_{r_1=1}^{M+1} \sum_{r_2=1}^{M-r_1+2} e^{-\frac{2\pi i}{N}\psi_0(r_2-1)} \int_0^{2\pi} \frac{d\phi}{2\pi} e^{-(r_1-1)\phi} \int_0^{2\pi} \frac{d\psi}{2\pi} e^{-(r_2-1)\psi}$$

$$\times \det_M \begin{vmatrix} s_F(\phi,\psi,1,1) & s_F(\phi,\psi,2,1) & \cdots & s_F(\phi,\psi,M,1) \\ s_F(\phi,\psi,1,2) & s_F(\phi,\psi,2,2) & \cdots & s_F(\phi,\psi,M,2) \\ \vdots & \vdots & \ddots & \vdots \\ s_F(\phi,\psi,1,M) & s_F(\phi,\psi,2,M) & \cdots & s_F(\phi,\psi,M,M) \end{vmatrix}, \quad (64)$$

where

$$s_F(\phi,\psi,l,j) = e^{i\phi} s_F^0(d_1,d_2,j,l) + e^{i\psi} s_F^1(d_1,d_2,j,l) + s_F^2(d_1,d_2,j,l). \quad (65)$$

Because the determinant appearing in (64) is of dimension $M$ we have $\det[\phi,\psi] = \sum_{m=0}^{M} \sum_{n=0}^{M} f_{m,n} e^{im\phi} e^{in\psi}$. Using this expression in (64) we obtain our final expression for the fermionic pseudo-spin function

$$S_F(d_1,d_2) = (N-M-1)!M!(-1)^{d_2-d_1} \sum_{r_1=1}^{M+1} \sum_{r_2=1}^{M-r_1+2} e^{-\frac{2\pi i}{N}\psi_0(r_2-1)} f_{r_1-1,r_2-1}, \quad (66)$$

which is Eq. (17) of Sec. III.

## V. TIME EVOLUTION OF THE CORRELATORS

We want to investigate the dynamics of the BFM in two experimentally relevant non-equilibrium situations: release from the trap and a sudden change of the trap frequency. In both cases we will deal with the following quench scenario: the system is prepared in the LL ground state of the Hamiltonian (1) with $\omega(t \le 0) = \omega_0$ and we suddenly change the trap frequency to a new value $\omega(t > 0) = \omega_1$ (when $\omega_1 = 0$ we have free expansion). The time evolution of the system will be governed by (1) with the new frequency.

If we denote by $|\Phi_{N,M}^{\omega_0}(\boldsymbol{j},\boldsymbol{\lambda})\rangle$ the ground-state of the pre-quench Hamiltonian, a general method of investigating the dynamics is to expand this state in the eigenstates of the of the post-quench Hamiltonian, which we will denote by $|\Phi_{N,M}^{\omega_1}(\boldsymbol{j'},\boldsymbol{\lambda'})\rangle$, and then apply the time-evolution operator (see [17] for the Tonks-Girardeau gas). However, in the case of multi-component systems the presence of the pseudo-spin degrees of freedom makes the application of this general method almost impossible. Now, we make two observations which will allow for the analytical and numerical investigation of the dynamics. The first observation is that the eigenstates of both Hamiltonians are described by the same formulae (2), (3) and (5) the only difference being that the Slater determinants describing the charge degrees of freedom contain Hermite functions of frequency $\omega_0$ ($\omega_1$) for the pre-quench(post-quench) Hamiltonian. The second observation and the most important is that due to the product form of the wavefunctions and completeness of both

charge and pseudo-spin wavefunctions the pseudo-spin structure of the initial state remains unchanged during time evolution. This can be seen easily by expanding the pre-quench state in the eigenstates of the post-quench Hamiltonian obtaining (see Appendix B)

$$
\begin{aligned}
|\Phi_{N,M}^{\omega_0}(\boldsymbol{j},\boldsymbol{\lambda})\rangle &= \sum_{\boldsymbol{j'},\boldsymbol{\lambda'}} |\Phi_{N,M}^{\omega_1}(\boldsymbol{j'},\boldsymbol{\lambda'})\rangle\langle\Phi_{N,M}^{\omega_1}(\boldsymbol{j'},\boldsymbol{\lambda'})|\Phi_{N,M}^{\omega_0}(\boldsymbol{j},\boldsymbol{\lambda})\rangle \\
&= \sum_{\boldsymbol{j'},\boldsymbol{\lambda'}} \delta_{\boldsymbol{\lambda},\boldsymbol{\lambda'}} B(\boldsymbol{j},\boldsymbol{j'})|\Phi_{N,M}^{\omega_1}(\boldsymbol{j'},\boldsymbol{\lambda'})\rangle = \sum_{\boldsymbol{j'}} B(\boldsymbol{j},\boldsymbol{j'})|\Phi_{N,M}^{\omega_1}(\boldsymbol{j'},\boldsymbol{\lambda})\rangle\,,
\end{aligned}
\tag{67}
$$

with $B(\boldsymbol{j},\boldsymbol{j'})$ defined in (B10) result which shows that the pseudo-spin configuration is frozen during the dynamics. Taking into account this observation and the factorized form of the wavefunction (3) this means that the time dependence of the system is encoded in the charge degrees of freedom. The time evolution of the harmonic oscillator with variable frequency is well known [77, 78] and is given by the scaling transformation

$$
\phi_j(x,t) = \frac{1}{\sqrt{b}}\phi_j\left(\frac{x}{b},0\right)\exp\left[i\frac{mx^2}{2\hbar}\frac{\dot{b}}{b} - iE_j\tau(t)\right]\,,
\tag{68}
$$

where $b(t)$ is a solution of the Ermakov-Pinney equation $\ddot{b} = -\omega(t)^2 b + \omega_0^2/b^3$ with boundary conditions $b(0) = 1$, $\dot{b}(0) = 0$, $E_j = \hbar\omega_0(j + 1/2)$ and the rescaled time parameter is given by $\tau(t) = \int_0^t dt'/b^2(t')$. From (3) and (12) we see that the dynamics of the correlators is given by

$$
g_\sigma(\xi_1,\xi_2;t) = \frac{1}{b}g_\sigma\left(\frac{\xi_1}{b},\frac{\xi_2}{b};0\right)e^{-\frac{i}{b}\frac{\dot{b}}{\omega_0}\frac{\xi_1^2-\xi_2^2}{2l_{HO}^2}}\,.
\tag{69}
$$

The evolution of the densities is given by $\rho_\sigma(\xi,t) \equiv g_\sigma(\xi,\xi;t) = \rho_\sigma(\xi/b)/b$ and the momentum distribution satisfies

$$
n_\sigma(p,t) = \frac{b}{2\pi}\int g_\sigma(\xi_1,\xi_2;0)e^{-ib\left[\frac{\dot{b}}{\omega_0}\frac{\xi_1^2-\xi_2^2}{2l_{HO}^2} - \frac{p(\xi_1-\xi_2)}{\hbar}\right]}d\xi_1 d\xi_2\,.
\tag{70}
$$

Formulae (69) and (70) show that in order to investigate the dynamics of the BFM after a sudden change of the trap frequency we only need to know the correlation functions at $t = 0$ which were presented in Sec. III and the solution of the Ermakov-Pinney equation. We should point out that these results remain valid in other quench scenarios like the case of a sinusoidal modulation of the trap frequency of the form $\omega^2(t) = \omega_0^2(1 - \alpha\sin\Omega t)$ for $t > 0$ with $\alpha,\Omega$ arbitrary parameters.

In Ref. [57] the authors proved the existence of dynamical fermionization in multi-component Bose and Fermi gases *assuming* that the spin degrees of freedom remain frozen during the expansion. The results of this section can be easily extended to prove this assumption in the case of multi-component Bose or Fermi gases the only necessary ingredients being the (pseudo)spin-charge factorization of the full wavefunctions and the completeness of the (pseudo)-spin and charge wavefunctions. For example, in the case of the bosonic or fermionic trapped Gaudin-Yang model [79, 80] the eigenstates and wavefunctions have the same form as (2) and (3) (we identify the spin-down particles with bosons and spin-up particles with fermions) but in this case the spin sector can be described by the XX0 spin-chain [50] (equivalent to hard-core bosons on the lattice) or the XXX spin-chain. Then, the proof that the spin configuration remains frozen during expansion is similar with the derivation of (67) and the results of Appendix B.

## VI. ANALYTICAL DERIVATION OF DYNAMICAL FERMIONIZATION

Using the results for the correlators presented in Sec. III and Eqs. (69) and (70) we can investigate both analytically and numerically the dynamics of a BFM after release from the trap along the same lines like in the case of the Tonks-Girardeau gas [38]. Even though the dynamics is encoded in the charge degrees of freedom the pseudo-spin structure of the initial ground-state plays an important role influencing the momentum distribution of each component of the mixture during the time-evolution. Compared with the analysis of [57] because we use an explicit form of the wavefunctions (3) we will be able to analytically investigate the densities and momentum distributions of both types of particles and not only their sums.

When the trap is released and the system evolves freely we have $\omega_1 = 0$ and the solution of the Ermakov-Pinney equation is $b(t) = \left(1 + \omega_0^2 t^2\right)^{1/2}$. We will derive the three properties mentioned in the introduction starting with the

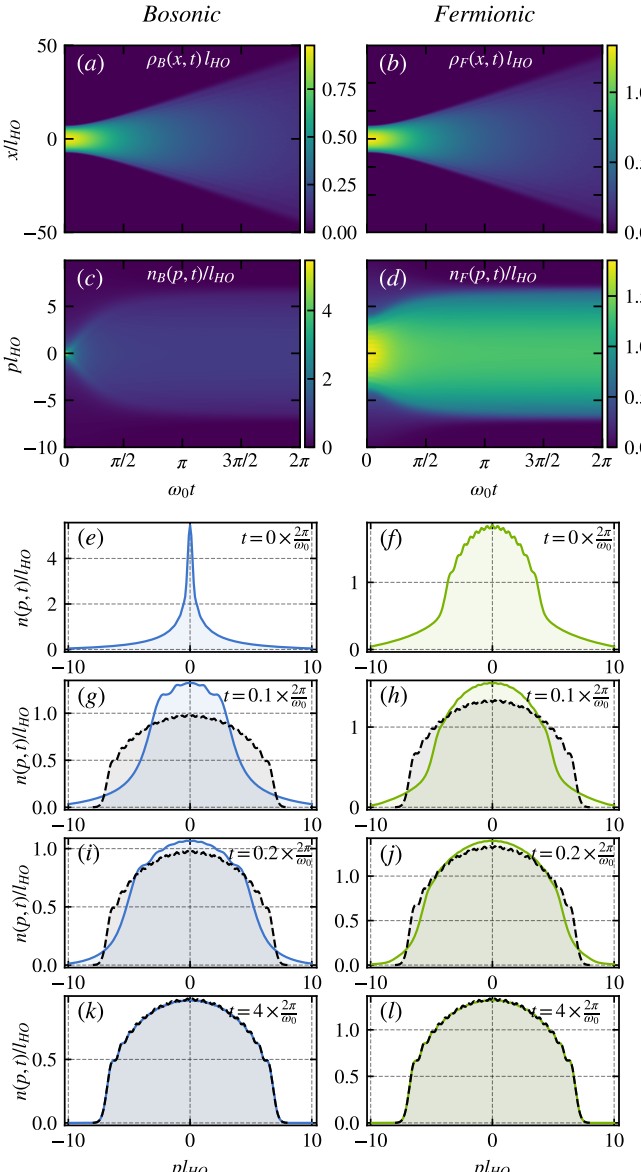

FIG. 1. Expansion and dynamical fermionization of a BFM with $N = 26$ and $M = 11$. Here $\omega_0 = 6$ and $l_{HO} = 1$. (a), (b) Evolution of the real-space densities $\rho_{B,F}(x,t)l_{HO}$ and (c), (d) momentum distributions $n_{B,F}(p,t)/l_{HO}$ as functions of the dimensionless time $\omega_0 t$. (e)–(g) Momentum distributions for $\omega_0 t = \{0, 0.1, 0.2, 4\} \times 2\pi$. In the initial LL ground-state the bosonic momentum distribution (e) presents quasi-condensate features (significant fraction of particles with $n_B(p) \sim 0$) while $n_F(p)$ (f) is narrower than the momentum distribution of trapped free fermions. In the last three rows the dashed lines represent the predictions of Eq. 74.

determination of the mixture's densities at $t = 0$. When $\xi_1 = \xi_2 = \xi$ Eq. (13) takes the form (see Appendix C)

$$\rho_\sigma(\xi) \equiv g_\sigma(\xi, \xi) = \sum_{d=1}^{N} \frac{1}{c_\sigma N^M} S_B(d, d) I(d, d; \xi, \xi), \tag{71}$$

and it is shown in Appendix D that $S_B(d, d) = (N - M)!M!N^{M-1}$ and $S_F(d, d) = (N - M - 1)!M!N^{M-1}(N - M)$. From Appendix C we can see that $\sum_{d=1}^{N} I(d, d; \xi, \xi) = \rho_{FF}(\xi)$ with $\rho_{FF}(\xi) = \sum_{j=0}^{N-1} \bar{\phi}_j(\xi)\phi_j(\xi)$ the density of a system of $N$ free fermions in the initial trap. These identities are independent of $\boldsymbol{\lambda}$ which means that the results derived below are also valid in the SILL regime. From these previous relations we find (this is the fermionization of

the mixture)

$$\rho_B(\xi) = \frac{M}{N}\rho_{FF}(\xi)\,,\ \ \rho_F(\xi) = \frac{N-M}{N}\rho_{FF}(\xi)\,,\ \ \rho_B(\xi) + \rho_F(\xi) = \rho_{FF}(\xi)\,,\ \ \ \ \ (72)$$

which shows that at $t = 0$ the densities of the BFM are proportional to the density of spinless free fermions (Property 0). In the large $t$ limit the integrals appearing in (70) can be investigated using the method of stationary phase (Chap. 6 of [81] or Chap. 2.9 of [82]). For both integrals the points of stationary phase are $\xi_0 = p\omega_0 l_{HO}^2/(\dot{b}\hbar)$ and we find

$$n_\sigma(p,t) \underset{t\to\infty}{\sim} \left|\frac{\omega_0 l_{HO}^2}{\dot{b}}\right| g_\sigma\left(\frac{p\omega_0 l_{HO}^2}{\dot{b}\hbar}, \frac{p\omega_0 l_{HO}^2}{\dot{b}\hbar}; 0\right)\,.\ \ \ \ \ (73)$$

Using $\lim_{t\to\infty} b(t) = \omega_0 t$, $\lim_{t\to\infty} \dot{b}(t) = \omega_0$ the last relation can be rewritten as $n_\sigma(p,t) \underset{t\to\infty}{\sim} l_{HO}^2 \rho_\sigma\left(pl_{HO}^2/\hbar, 0\right)$ proving that the asymptotic momentum distribution of each component has the same shape as its initial real space density (Property 1) [in $\hbar = m = \omega_0 = 1$ units this takes the form $n_\sigma(p,t) \underset{t\to\infty}{\sim} \rho_\sigma(p,0)$]. Using the results for the initial densities and the identity $n_{FF}(p) = l_{HO}^2 \rho_{FF}\left(pl_{HO}^2/\hbar\right)$ (see Appendix E) where $n_{FF}(p)$ is the momentum distribution of spinless noninteracting fermions we find that in the large $t$ limit

$$n_B(p,t) \sim \frac{M}{N}n_{FF}(p)\,,\ \ n_F(p,t) \sim \frac{N-M}{N}n_{FF}(p)\,,\ \ \ \ n_B(p,t) + n_F(p,t) \sim n_{FF}(p)\,,\ \ \ \ \ (74)$$

which proves the dynamical fermionization of the mixture (Property 2). The dynamics after release from the trap of a system of $N = 26$ particles of which $M = 11$ are bosons is shown in Fig. 1. The bosonic momentum distribution in the LL ground-state which can be seen in Fig. 1e) presents similar quasi-condensate characteristics like in the case of the Tonks-Girardeau gas but we should point out that while for the homogeneous TG gas $n_{TG}(p) \sim |p|^{-1/2}$ in the case of the homogeneous BFM mixture we have $n_B(p) \sim |p|^{-1+1/(2K_B)}$ with $K_B = 1/[(1 - M/N)^2 + 1]$ [83]. The presence of the harmonic trap smoothens this singularity significantly. The initial fermionic distribution [Fig. 1(f)] presents $N - M$ local maxima (similar to the free fermionic case) but is narrower than the momentum distribution of a similar number of free fermions subjected to the same harmonic trapping. At large momenta both the bosonic and fermionic momentum distributions behave like $\lim_{p\to\pm\infty} n_{B,F}(p) \sim C_{B,F}/p^4$ with $C_{B,F}$ the Tan contacts [84–92]. After the harmonic potential is removed the system undergoes dynamical fermionization Fig. 1 (g)-(l) with the asymptotic momentum distributions being described by Eq. (74) (dashed lines in the last three rows of Fig. 1).

## VII. BREATHING OSCILLATIONS

A sudden change of the trap frequency to a nonzero value $\omega_1$ induces in the system the so-called breathing oscillations, which can have large amplitudes when $\omega_0/\omega_1 \gg 1$ and were experimentally investigated in the case of the Lieb-Liniger model in [28, 45, 46]. In this case the solution of the Ermakov-Pinney equation is $b(t) = \left[1 + (\omega_0^2 - \omega_1^2)\sin^2(\omega_1 t)/\omega_1^2\right]^{1/2}$ which is periodic with period $T = \pi/\omega_1$ and takes values between 1 and $\omega_0/\omega_1$ (note that the solution for the expansion can be obtained in the limit $\omega_1 \to 0$).

While the dynamics of the real-space density (see Eq. (69)) consists of self-similar breathing cycles with no damping, Fig. 2 (a)-(b), the dynamics in momentum space presents a richer structure represented by bosonic-fermionic oscillations as it can be seen in Fig. 2 (c)-(h). Let us focus first on the time-evolution of the bosonic momentum distribution. Similar to the case of the TG gas which was investigated in [22, 23] the bosonic momentum distribution of the Bose-Fermi mixture presents periodic narrowings which occurs at twice the rate of the density's oscillations. This phenomenon is absent in the case of free spinless fermions in the trap and was interpreted as a quantum many-body bounce effect [22, 23]. In order to get a clearer picture of this phenomenon in Fig. 3 we plot the time evolution of the momentum distribution's Full Width at Half Maximum (FWHM) of a system with $N = 18$ particles and different number of bosons. The results for the FWHM of $n_B(p)$ are shown in Fig. 3(a), (c) and (e) (blue continuous lines) along the similar results for the FWHM of a TG gas with the same number of bosons undergoing the same quench (dashed black lines).

The bosonic FWHM dynamics is very similar with the case of single-component TG bosons [38] and is characterized by a rapid initial increase which can be understood as a partial dynamical fermionization due to the fact that $\omega_0 > \omega_1$. What is not necessarily intuitive is the presence of local minima at $t = mT/2$ ($m$ integer) revealing a distribution which is narrower and taller than the one at $t = 0$ (for the TG gas $n(p; T/2) = (\omega_0/\omega_1)n(\omega_0 p/\omega_1; 0)$) and the fact that the FWHM has additional minima at $t = mT$ where the real space density is the narrowest feature which is not present in a noninteracting system. We should also point out that the periodic narrowing occurring at $t = mT$ takes

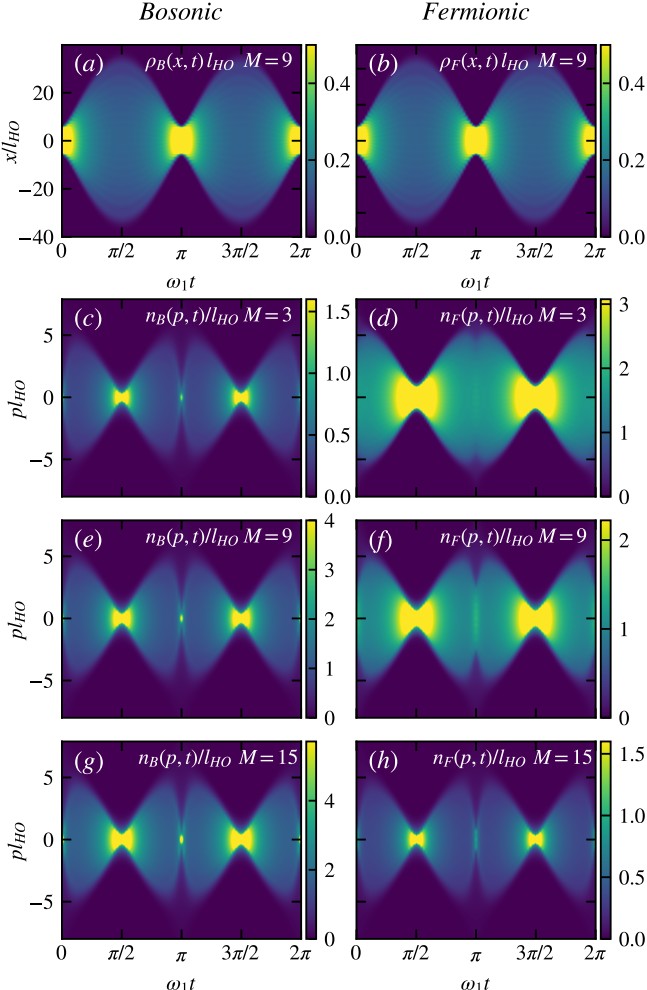

FIG. 2. Dynamics of the BFM after a confinement quench with $\omega_0 = 6$ ($l_{HO} = 1$) and $\omega_1 = 1$. (a), (b) Evolution of the real-space densities $\rho_{B,F}(x,t)l_{HO}$ as functions of the dimensionless time $\omega_1 t$ for $N = 18$ and $M = 9$. Different number of bosons do not change this picture significantly. Evolution of the momentum distributions $n_{B,F}(p,t)/l_{HO}$ for $N = 18$ and $M = 3$ (c), (d) $M = 9$ (e), (f) and $M = 15$ (g), (h). In the bosonic case note the periodic narrowing of the momentum distribution occurring at twice the rate of the density oscillations. In the fermionic case the narrowing at $t = mT$ is strongly influenced by the number of bosons in the system and it eventually disappears for a purely fermionic system ($M = 0$).

place on a relatively short time scale compared with the one occurring at $t = mT/2$. The bosonic FWHM of the BFM presents the same features with two particularities: a) the FWHM at $t = mT$ is narrower than the FWHM of a TG gas with the same number of bosons and b) the FWHM at all $t$ is almost independent on the number of bosons and depends only on the total number of particles $N$ of the mixture.

The dynamics of the fermionic momentum distribution presents significant changes compared to the case of spinless free fermions subjected to the same quench and can be seen in Fig. 3 (b), (d) and (f) (the continuous green lines are the momentum distribution's FWHM of the BFM fermions while the dashed black lines represent the momentum distribution's FWHM of a system of $N - M$ free fermions undergoing the same quench). As a result of the interaction between the bosonic and fermionic particles the fermionic momentum distribution of the BFM at $t = mT$ is narrower than the momentum distribution of a similar number of free fermions and this narrowing becomes more pronounced as the number of bosons in the mixture increases. Therefore, while in the free fermionic case the FWHM presents maxima at $t = mT$ and minima at $t = mT/2$ and is monotonic in between the fermionic momentum distribution of the mixture presents features which are similar with the bosonic one and depends heavily on the bosonic fraction $M/N$ of the system. Not only the fermionic momentum distribution of the interacting system becomes significantly narrower at $t = mT$ (which are local minima) but it also presents local maxima at $t \neq mT$ similar to the bosonic case. It is interesting to note that while after release from the trap the system effectively "fermionizes" in the case of

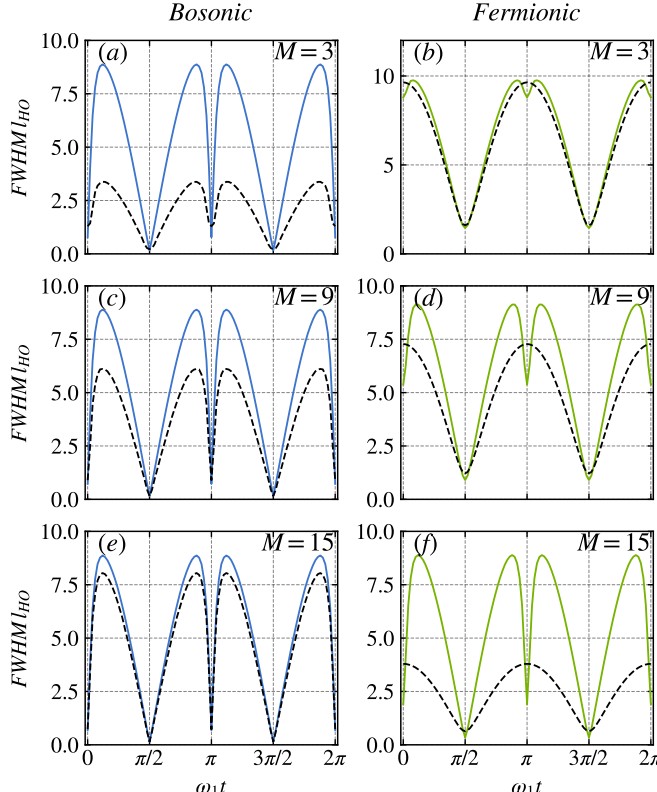

FIG. 3. FWHM dynamics of the momentum distributions (blue continuous lines for bosons and green continuous lines for fermions) after a confinement quench with the same trap parameters as in Fig. 2 for $N = 18$ and $M = \{3, 9, 15\}$. In (a), (c) and (e) the dashed black lines represent the FWHM dynamics of the momentum distribution of $M$ Tonks-Girardeau bosons subjected to the same confinement quench. In (b), (d) and (f) the dashed black lines represent the FWHM dynamics of the momentum distribution of $N - M$ free fermions in the same quench scenario.

the breathing oscillations initiated by a sudden change of the trap frequency the dynamics (especially in the case of the full momentum distribution) is similar to the one of the TG gas phenomenon which can be dubbed "dynamical bosonization".

## VIII. CONCLUSIONS

In this article we have derived an analytical description for the correlation functions of the Bose-Fermi mixture in a harmonic trap which can be easily implemented numerically with a complexity which is polynomial in the number of particles. Our results represent a substantial improvement over other numerical approaches of trapped multi-component systems which have an exponential complexity in the number of particles or require hybrid techniques in which the charge degrees of freedom are examined using analytical methods while the spin sector is investigated with the infinite time-evolving block decimation (iTEBD) or the Density Matrix Renormalization Group (DMRG) algorithms. Using the product form of the wavefunctions we have demonstrated that in various non-equilibrium scenarios the pseudo-spin sector of the BFM remains frozen with the dynamics being encoded in the charge degrees of freedom. Our proof can be easily generalized to any strongly interacting gas with an arbitrary number of components and any statistics of the particles. We have also proved, analytically and numerically that dynamical fermionization occurs in the BFM after the release from the trap and that the asymptotic momentum distribution of each component approaches the shape of their initial real space densities. In the case of a quench of the trap frequency which induces breathing oscillations we have shown that contrary to usual expectations, the system exhibits features which are similar to a system of single component bosons and not that of a system of polarized fermion subjected to a similar quench. In this case we can say that the system effectively exhibits a sort of "dynamical bosonization" in contrast with the case of free expansion.

Following the arguments of [57] it can be argued that in the case of the expansion of the system some of the features

derived in Sec.VI remain valid in the case of strong but finite interaction. In this case the full wavefunctions (to leading order) are still of product form (see [93]) with the charge degrees of freedom described by Slater determinants of Hermite functions but with the spin sector described by the wavefunctions of an XXZ spin-chain [56] with variable exchange coefficients $C_i$ . What is remarkable is that in the case of harmonic trapping the time evolution of these coefficients is given by $C_i(t) = C_i(0)b^{-3}(t)$ [94] which results in an overall scaling of the spin Hamiltonian. Therefore, an eigenstate of the spin Hamiltonian at $t = 0$ remains an eigenstate at $t > 0$ resulting in an frozen spin configuration like in the impenetrable case and then the same logic applies like in Sec.VI. The results obtained in this paper reveal the interesting phenomena resulting from the interplay between the strong interaction, internal degrees of freedom and statistics which can be probed using current ultracold gases experiments.

**ACKNOWLEDGMENTS**

Financial support from the LAPLAS 6 program of the Romanian National Authority for Scientific Research (CNCS-UEFISCDI) is gratefully acknowledged.

**Appendix A: Examples of the wavefunctions for $N = 4$ and $M = 2$**

A distinct feature of the BFM wavefunction compared with the case of the bosonic or fermionic Gaudin-Yang models [50] is that the ordering of the set of integers $\boldsymbol{n} = (n_1, \cdots, n_M)$, $n_a \in \{1, \cdots, N\}$, describing the positions of the bosons in the ordered set $\{x_1, \cdots, x_N\}$ is relevant. If we consider $N = 4$, $M = 2$, $\boldsymbol{\alpha} = (BFBF)$ and the wedge $\theta(x_1 < x_2 < x_3 < x_4)$ then $\boldsymbol{n} = (1, 3)$. For the permutation $P = (3214)$ the pseudo-spin wavefunction in the sector $\theta(x_3 < x_2 < x_1 < x_4)$ is $\det_M \left( e^{in_a \lambda_b} \right)$ with $\boldsymbol{n} = (3, 1)$ because the first boson is in the third position of the ordered set while the second boson is in the first position. Let us present the full wavefunctions for $N = 4$, $M = 2$ and two sets of $\boldsymbol{\alpha}$. The full expressions are rather long so introduce the notations $(jklm) = \theta(x_j < x_k < x_l < x_m)$ and $[jk] = e^{i(j\lambda_1 + k\lambda_2)} - e^{i(k\lambda_1 + j\lambda_2)}$. First, we consider the case $\boldsymbol{\alpha} = (BBFF)$ which describes a system in which the first (second) boson has the coordinate $x_1(x_2)$. The wavefunction is

$$
\begin{aligned}
\chi^{BBFF}(x_1, x_2, x_3, x_4 | \boldsymbol{j}, \boldsymbol{\lambda}) = &\frac{\det_4 [\phi_{j_a}(x_b)]}{4 \times 4!} \\
&\times \{ \ [12](1234) + [12](1243) + [13](1324) + [14](1342) + [13](1423) + [14](1432) \\
&\quad + [21](2134) + [21](2143) + [31](2314) + [41](2341) + [31](2413) + [41](2431) \\
&\quad + [23](3124) + [24](3142) + [32](3214) + [42](3241) + [34](3412) + [43](3421) \\
&\quad + [23](4123) + [24](4132) + [32](4213) + [42](4231) + [34](4312) + [43](4321) \ \} \ .
\end{aligned}
$$

When $\boldsymbol{\alpha} = (BFBF)$ the wavefunction is

$$
\begin{aligned}
\chi^{BFBF}(x_1, x_2, x_3, x_4 | \boldsymbol{j}, \boldsymbol{\lambda}) = &\frac{\det_4 [\phi_{j_a}(x_b)]}{4 \times 4!} \\
&\times \{ \ [13](1234) + [14](1243) + [12](1324) + [12](1342) + [14](1423) + [13](1432) \\
&\quad + [23](2134) + [24](2143) + [32](2314) + [42](2341) + [34](2413) + [43](2431) \\
&\quad + [21](3124) + [21](3142) + [31](3214) + [41](3241) + [31](3412) + [41](3421) \\
&\quad + [24](4123) + [23](4132) + [34](4213) + [43](4231) + [32](4312) + [42](4321) \ \} \ .
\end{aligned}
$$

It is now easy to see that $\chi^{BBFF}(x_1, x_2, x_3, x_4) = -\chi^{BFBF}(x_1, x_3, x_2, x_4)$ which is just a particular case of the generalized symmetry

$$
\chi^{\alpha_1 \cdots \alpha_i \alpha_{i+1} \cdots \alpha_N}(x_1, \cdots, x_i, x_{i+1}, \cdots, x_N) = h_{\alpha_i \alpha_{i+1}} \chi^{\alpha_1 \cdots \alpha_{i+1} \alpha_i \cdots \alpha_N}(x_1, \cdots, x_{i+1}, x_i \cdots, x_N) , \tag{A1}
$$

satisfied by the wavefunctions [the exchange of $x_2$ with $x_3$ means for example that $(1342)$ transforms in $(1243)$].

**Appendix B: Orthogonality and normalization of the eigenstates**

Here we will prove that the eigenstates defined in Sec. II are orthogonal and normalized. The eigenstates of different $(N, M)$-sectors are obviously orthogonal so we will focus on computing

$$
A_{(N,M)} \equiv \langle \Phi_{N,M}(\boldsymbol{j'}, \boldsymbol{\lambda'}) | \Phi_{N,M}(\boldsymbol{j}, \boldsymbol{\lambda}) \rangle ,
$$

$$= \int \prod_{j=1}^{N} dx_j dy_j \sum_{\alpha_1,\cdots,\alpha_N=\{B,F\}}^{[N,M]} \sum_{\beta_1,\cdots,\beta_N=\{B,F\}}^{[N,M]} \bar{\chi}_{N,M}^{\alpha_1\cdots\alpha_N}(\boldsymbol{x}|\boldsymbol{j}',\boldsymbol{\lambda}')\chi_{N,M}^{\beta_1\cdots\beta_N}(\boldsymbol{y}|\boldsymbol{j},\boldsymbol{\lambda})$$
$$\times \langle 0|\Psi_{\beta_1}(y_1)\cdots\Psi_{\beta_N}(y_N)\Psi_{\alpha_N}^\dagger(x_N)\cdots\Psi_{\alpha_1}^\dagger(x_1)|0\rangle, \qquad (B1)$$

where we have introduced $\boldsymbol{x}=(x_1,\cdots,x_N)$ and $\boldsymbol{y}=(y_1,\cdots,y_N)$. This is a rather daunting expression so it is useful to consider some particular cases. In the case of the $(2,1)$-sector repeated applications of the commutation relations gives for the part involving the fields

$$\langle 0|\Psi_{\beta_1}(y_1)\Psi_{\beta_2}(y_2)\Psi_{\alpha_2}^\dagger(x_2)\Psi_{\alpha_1}^\dagger(x_1)|0\rangle = h_{\alpha_2\beta_2}\delta_{\alpha_1\beta_2}\delta_{\alpha_2\beta_1}\delta(x_1-y_2)\delta(x_2-y_1) + \delta_{\alpha_1\beta_1}\delta_{\alpha_2\beta_2}\delta(x_1-y_1)\delta(x_2-y_2) \quad (B2)$$

and, therefore,

$$A_{(2,1)} = \int dx_1 dx_2 \sum_{\alpha_1,\alpha_2=\{B,F\}}^{[2,1]} \left[ h_{\alpha_2\alpha_1}\bar{\chi}_{2,1}^{\alpha_2\alpha_1}(x_2,x_1|\boldsymbol{j}',\boldsymbol{\lambda}')\chi_{2,1}^{\alpha_1\alpha_2}(x_1,x_2|\boldsymbol{j},\boldsymbol{\lambda}) + \bar{\chi}_{1,2}^{\alpha_1\alpha_2}(x_1,x_2|\boldsymbol{j}',\boldsymbol{\lambda}')\chi_{2,1}^{\alpha_1\alpha_2}(x_1,x_2|\boldsymbol{j},\boldsymbol{\lambda}) \right]$$
$$=2 \int dx_1 dx_2 \sum_{\alpha_1,\alpha_2=\{B,F\}}^{[2,1]} \bar{\chi}_{2,1}^{\alpha_1\alpha_2}(x_1,x_2|\boldsymbol{j}',\boldsymbol{\lambda}')\chi_{2,1}^{\alpha_1\alpha_2}(x_1,x_2|\boldsymbol{j},\boldsymbol{\lambda}), \qquad (B3)$$

where in the last line we have used the generalized symmetry (A1). The generalization of (B3) in the $(N,M)$-sector is

$$A_{(N,M)} = N! \int \prod_{j=1}^{N} dx_j \sum_{\alpha_1,\cdots,\alpha_N=\{B,F\}}^{[N,M]} \bar{\chi}_{N,M}^{\alpha_1\cdots\alpha_N}(\boldsymbol{x}|\boldsymbol{j}',\boldsymbol{\lambda}')\chi_{N,M}^{\alpha_1\cdots\alpha_N}(\boldsymbol{x}|\boldsymbol{j},\boldsymbol{\lambda}). \qquad (B4)$$

Using the expression for the wavefunctions (3) we get

$$A_{(N,M)} = \frac{1}{N!N^M} \int \prod_{j=1}^{N} dx_j \sum_{\alpha_1,\cdots,\alpha_N=\{B,F\}}^{[N,M]} \left[ \sum_{P\in S_N} \bar{\eta}_{N,M}^{\alpha_{P_1}\cdots\alpha_{P_N}}(\boldsymbol{\lambda}')\theta(P\boldsymbol{x}) \right] \left[ \sum_{Q\in S_N} \eta_{N,M}^{\alpha_{Q_1}\cdots\alpha_{Q_N}}(\boldsymbol{\lambda})\theta(Q\boldsymbol{x}) \right]$$
$$\times \det_N \left[ \bar{\phi}_{j'_a}(x_b) \right] \det_N \left[ \phi_{j_a}(x_b) \right]. \qquad (B5)$$

Now we need to make two important observations. The first observation is that $\theta(P\boldsymbol{x})\theta(Q\boldsymbol{x}) = \theta(P\boldsymbol{x})\delta_{PQ}$ for arbitrary permutations $P$ and $Q$ and the second observation is that for any permutation $P$ the sum $\sum_{P\in S_N} \bar{\eta}_{N,M}^{\alpha_{P_1}\cdots\alpha_{P_N}}(\boldsymbol{\lambda}')\eta_{N,M}^{\alpha_{P_1}\cdots\alpha_{P_N}}(\boldsymbol{\lambda})$ is the same due to the fact that the spin wavefunctions involve determinants and therefore their product is symmetric in $n$'s. Using these observations we obtain

$$A_{(N,M)} = \frac{1}{N!N^M} \int \prod_{j=1}^{N} dx_j \left[ \sum_{\alpha_1,\cdots,\alpha_N=\{B,F\}}^{[N,M]} \bar{\eta}_{N,M}^{\alpha_{P_1}\cdots\alpha_{P_N}}(\boldsymbol{\lambda}')\eta_{N,M}^{\alpha_{P_1}\cdots\alpha_{P_N}}(\boldsymbol{\lambda}) \right] \sum_{P\in S_N} \theta(P\boldsymbol{x})$$
$$\times \det_N \left[ \bar{\phi}_{j'_a}(x_b) \right] \det_N \left[ \phi_{j_a}(x_b) \right]. \qquad (B6)$$

We focus now on the term in the square parenthesis. Because the product $\bar{\eta}\eta$ is symmetric in $n$'s and vanishes when two of them are equal we have

$$\left[ \sum_{\alpha_1,\cdots,\alpha_N=\{B,F\}}^{[N,M]} \bar{\eta}_{N,M}^{\alpha_{P_1}\cdots\alpha_{P_N}}(\boldsymbol{\lambda}')\eta_{N,M}^{\alpha_{P_1}\cdots\alpha_{P_N}}(\boldsymbol{\lambda}) \right] = \frac{1}{M!} \sum_{n_1=1}^{N} \cdots \sum_{n_M=1}^{N} \det_M \left( e^{-i\lambda'_a n_b} \right) \det_M \left( e^{i\lambda_a n_b} \right) \qquad (B7)$$

with the $1/M!$ factor appearing because for a symmetric function there are $M!$ permutations for a particular $\boldsymbol{n}$ which give the same result. Expanding the determinants from the right hand side of (B7) we obtain

$$r.h.s = \frac{1}{M!} \sum_{n_1=1}^{N} \cdots \sum_{n_M=1}^{N} \left( \sum_{P\in S_N} (-1)^P \prod_{j=1}^{M} e^{-in_j\lambda'_{P_j}} \right) \left( \sum_{Q\in S_N} (-1)^P \prod_{j=1}^{M} e^{in_j\lambda_{Q_j}} \right),$$

$$= \frac{1}{M!} \sum_{n_1=1}^{N} \cdots \sum_{n_M=1}^{N} \left( \sum_{P \in S_N} \sum_{Q \in S_N} (-1)^{P+Q} \prod_{j=1}^{M} e^{-in_j(\lambda'_{P_j} - \lambda_{Q_j})} \right) ,$$

$$= \frac{1}{M!} \sum_{P \in S_N} \sum_{Q \in S_N} (-1)^{P+Q} \prod_{j=1}^{M} \left( \sum_{n=1}^{N} e^{-in(\lambda'_{P_j} - \lambda_{Q_j})} \right) ,$$

$$= \frac{1}{M!} \sum_{P \in S_N} \sum_{Q \in S_N} (-1)^{P+Q} \prod_{j=1}^{M} N \delta_{\lambda'_{P_j} \lambda_{Q_j}} = N^M \delta_{\boldsymbol{\lambda'\lambda}} , \tag{B8}$$

where in the third line we have summed the geometric series and used the fact that $e^{i\lambda'_j N} = e^{i\lambda_j N} = 1$ for all $j \in \{1, \cdots, M\}$. Using this last relation in (B6) and that $\sum_{P \in S_N} \theta(P\boldsymbol{x})$ is the identity in $\mathbb{R}^N$ we have

$$A_{(N,M)} = \frac{\delta_{\boldsymbol{\lambda'\lambda}}}{N!} \int \prod_{j=1}^{N} dx_j \left( \sum_{P \in S_N} (-1)^P \prod_{k=1}^{N} \bar{\phi}_{j'_{P_k}}(x_k) \right) \left( \sum_{Q \in S_N} (-1)^Q \prod_{k=1}^{N} \phi_{j_{Q_k}}(x_k) \right) ,$$

$$= \frac{\delta_{\boldsymbol{\lambda'\lambda}}}{N!} \int \prod_{j=1}^{N} dx_j \sum_{P \in S_N} \sum_{Q \in S_N} (-1)^{P+Q} \prod_{k=1}^{N} \left( \bar{\phi}_{j'_{P_k}}(x_k) \phi_{j_{Q_k}}(x_k) \right) ,$$

$$= \frac{\delta_{\boldsymbol{\lambda'\lambda}}}{N!} \sum_{P \in S_N} \sum_{Q \in S_N} (-1)^{P+Q} \prod_{k=1}^{N} \delta_{j'_{P_k} j_{Q_k}} = \delta_{\boldsymbol{j'j}} \delta_{\boldsymbol{\lambda'\lambda}} , \tag{B9}$$

by using the orthonormality of the Hermite functions. This concludes the proof.

In the same way it can be shown that $\langle \Phi^{\omega_1}_{N,M}(\boldsymbol{j'}, \boldsymbol{\lambda'}) | \Phi^{\omega_0}_{N,M}(\boldsymbol{j}, \boldsymbol{\lambda}) \rangle = \delta_{\boldsymbol{\lambda},\boldsymbol{\lambda'}} B(\boldsymbol{j}, \boldsymbol{j'})$ with

$$B(\boldsymbol{j}, \boldsymbol{j'}) = \frac{1}{N!} \int \prod_{j=1}^{N} dx_j \sum_{P \in S_N} \sum_{Q \in S_N} (-1)^{P+Q} \prod_{k=1}^{N} \left( \bar{\phi}_{j'_{P_k}}(x_k; \omega_1) \phi_{j_{Q_k}}(x_k; \omega_0) \right) , \tag{B10}$$

where $\phi_j(x_k; \omega)$ is the Hermite function of frequency $\omega$, $|\Phi^{\omega_0}_{N,M}(\boldsymbol{j}, \boldsymbol{\lambda})\rangle$ is the ground-state of the pre-quench Hamiltonian and $|\Phi^{\omega_1}_{N,M}(\boldsymbol{j'}, \boldsymbol{\lambda'})\rangle$ an eigenstate of the post-quench Hamiltonian. In the case of the free expansion ($\omega_1 = 0$) the determinant $\det_N [\phi_{j_a}(x_b)]$ in the wavefunction is replaced by $\det_N [e^{ik_a x_b}]$ with $\{k_a\}$ satisfying the Bethe ansatz equations of the homogeneous system [63] but otherwise the same logic applies.

## Appendix C: Some relevant integrals in the new parametrization

The parametrization introduced in Sec. IV has the advantage of making explicit the factorization of the pseudo-spin and charge degrees of freedom in the definition of important physical quantities. In this appendix we present these factorizations for some relevant integrals including the expressions for the correlators (12) in an arbitrary state.

*Normalization integrals.* The simplest case is represented by the integrals that appear when calculating the normalization of wavefunctions such as (we drop unnecessary subscripts when there is no risk of confusion and we remind that $\boldsymbol{\alpha} = (B \cdots B F \cdots F)$)

$$A = \int \prod_{j=1}^{n} dx_j \, \bar{\chi}(x_1, \cdots, x_n) \chi(x_1, \cdots, x_n) . \tag{C1}$$

In the new parametrization we have ($c = 1/[(N!)^2 N^M]$)

$$A = c \sum_{\boldsymbol{y} \in S_N} \int_Z \prod_{j=1}^{n} dz_j \, \bar{\chi}(z_1, \cdots, z_n; \boldsymbol{y}) \chi(z_1, \cdots, z_n; \boldsymbol{y}) ,$$

$$= c \underbrace{\left[ \sum_{\boldsymbol{y} \in S_N} (-1)^{\boldsymbol{y}+\boldsymbol{y}} \det_M \left( e^{-iy_a \lambda_b} \right) \det_M \left( e^{iy_a \lambda_b} \right) \right]}_{S} \underbrace{\left[ \int_Z \prod_{j=1}^{n} dz_j \det_N \left[ \bar{\phi}_{j_a}(z_b) \right] \det_N \left[ \phi_{j_a}(z_b) \right] \right]}_{I} , \tag{C2}$$

which shows the factorization of the charge and spin degrees of freedom. In this case $I = 1$ and $S = (N - M)!M!N^M$.

*Densities integrals.* The following type of integrals appear in the expressions for the densities $\rho_\sigma(\xi) = g_\sigma(\xi, \xi)$. We have to treat the bosonic and fermionic cases independently. We start with the bosonic case. Introducing

$$Z_d(\xi) = \{-\infty \leq z_1 \leq \cdots \leq z_{d-1} \leq \xi \leq z_{d+1} \leq \cdots \leq z_N \leq +\infty\}, \tag{C3}$$

and

$$Y(y_1 = d) = \{\boldsymbol{y} \in S_N | \text{ with } y_1 = d\}, \tag{C4}$$

then the relevant integral is

$$
\begin{aligned}
A_B(\xi) &= \int_{\mathbb{R}^{N-1}} \prod_{j=2}^{N} dx_j \bar{\chi}(\xi, x_2, \cdots, x_N) \chi(\xi, x_2, \cdots, x_N), \\
&= c \sum_{d=1}^{N} \sum_{\boldsymbol{y} \in Y(y_1 = d)} \int_{Z_d(\xi)} \prod_{\substack{j=1, j \neq d}}^{N} dz_j \, \bar{\chi}(z_1, \cdots, \underset{\underset{d}{\uparrow}}{\xi}, \cdots, z_N; \boldsymbol{y}) \chi(z_1, \cdots, \underset{\underset{d}{\uparrow}}{\xi}, \cdots, z_N; \boldsymbol{y}), \\
&= c \sum_{d=1}^{N} \underbrace{\left[ \sum_{\boldsymbol{y} \in Y(y_1 = d)} (-1)^{\boldsymbol{y} + \boldsymbol{y}} \det_M \left( e^{-iy_a \lambda_b} \right) \det_M \left( e^{iy_a \lambda_b} \right) \right]}_{S_B(d)} \underbrace{\left[ \int_{Z_d(\xi)} \prod_{\substack{j=1, j \neq d}}^{N} dz_j \det_N \left[ \bar{\phi}_{j_a}(z_b) \right] \det_N \left[ \phi_{j_a}(z_b) \right] \right]}_{I(d; \xi)}, \\
&= c \sum_{d=1}^{N} S_B(d) I(d; \xi).
\end{aligned} \tag{C5}
$$

In the fermionic case we have

$$
\begin{aligned}
A_F(\xi) &= \int_{\mathbb{R}^{N-1}} \prod_{j=1}^{N-1} dx_j \bar{\chi}(x_1, \cdots, x_{N-1}, \xi) \chi(x_1, \cdots, x_{N-1}, \xi), \\
&= c \sum_{d=1}^{N} \sum_{\boldsymbol{y} \in Y(y_N = d)} \int_{Z_d(\xi)} \prod_{\substack{j=1, j \neq d}}^{N} dz_j \, \bar{\chi}(z_1, \cdots, \underset{\underset{d}{\uparrow}}{\xi}, \cdots, z_N; \boldsymbol{y}) \chi(z_1, \cdots, \underset{\underset{d}{\uparrow}}{\xi}, \cdots, z_N; \boldsymbol{y}), \\
&= c \sum_{d=1}^{N} \underbrace{\left[ \sum_{\boldsymbol{y} \in Y(y_N = d)} (-1)^{\boldsymbol{y} + \boldsymbol{y}} \det_M \left( e^{-iy_a \lambda_b} \right) \det_M \left( e^{iy_a \lambda_b} \right) \right]}_{S_F(d)} \underbrace{\left[ \int_{Z_d(\xi)} \prod_{\substack{j=1, j \neq d}}^{N} dz_j \det_N \left[ \bar{\phi}_{j_a}(z_b) \right] \det_N \left[ \phi_{j_a}(z_b) \right] \right]}_{I(d; \xi)}, \\
&= c \sum_{d=1}^{N} S_F(d) I(d; \xi),
\end{aligned} \tag{C6}
$$

where $Y(y_N = d)$ is defined analogously with (C4) but in this case $y_N = d$. From (C5) and (C6) we see again that the relevant integrals factorize and that while the pseudo-spin functions are different the charge integral is the same in both cases.

*Correlator integrals.* These are the integrals that appear in (12a) and (12b). We start with the bosonic integral

$$A_B(\xi_1, \xi_2) = \int_{\mathbb{R}^{N-1}} \prod_{j=2}^{N} dx_j \bar{\chi}(\xi_1, x_2, \cdots, x_N) \chi(\xi_2, x_2, \cdots, x_N). \tag{C7}$$

We will consider the case $\xi_1 \leq \xi_2$ and because in general $\xi_1 \neq \xi_2$ we have to introduce two sets of parameters for the two wavefunctions: $\boldsymbol{z}, \boldsymbol{y}$ for $\bar{\chi}$ and $\boldsymbol{z}', \boldsymbol{y}'$ for $\chi$. These two sets are not fully independent as we have $x_2' = x_2, \cdots, x_N' = x_N$. In order to see the connection it is useful to consider a particular case. Consider $(N, M) = (7, 3)$ and $d_1 = 3$ and $d_2 = 5$. Now we consider one of the wedges in which $\xi_1$ is on the $d_1$-th position in the ordered set $\xi_1, x_2, \cdots, x_N$ and $\xi_2$ is on the $d_2$-th position in the ordered set $\xi_2, x_2, \cdots, x_N$. If we consider the particular wedges

$$X = \{-\infty \leq x_2 \leq x_3 \leq \xi_1 \leq x_4 \leq x_5 \leq x_6 \leq x_7 \leq +\infty\},$$

$$X' = \{-\infty \le x_2 \le x_3 \le x_4 \le x_5 \le \xi_2 \le x_6 \le x_7 \le +\infty\}$$

then, in $z$ variables they correspond to

$$
\begin{aligned}
Z_{d_1}(\xi_1) &= \{-\infty \le z_1 \le z_2 \le \xi_1 \le z_4 \le z_5 \le z_6 \le z_7 \le +\infty\}\,, \\
Z'_{d_2}(\xi_2) &= \{-\infty \le z'_1 \le z'_2 \le z'_3 \le z'_4 \le \xi_2 \le z'_6 \le z'_7 \le +\infty\} = \{-\infty \le z_1 \le z_2 \le z_4 \le z_5 \le \xi_2 \le z_6 \le z_7 \le +\infty\}\,.
\end{aligned}
$$
$$\text{(C8)}$$

The $y$ parameters are $\boldsymbol{y} = (3,1,2,4,5,6,7)$ and $\boldsymbol{y'} = (5,1,2,3,4,6,7)$. In the general case the connection between the $y$ parameters is the following $(d_1 \le d_2)$:

$$
\begin{aligned}
y'_1 &= d_2\,, \quad y_1 = d_1\,, \\
y'_i &= y_i \quad \text{for } y_i < d_1\,, \\
y'_i &= y_i - 1 \quad \text{for } d_1 < y_i \le d_2\,, \\
y'_i &= y_i\,, \quad \text{for } d_2 < y_i\,.
\end{aligned}
$$
$$\text{(C9)}$$

while for the $z$ parameters we have the constraint

$$-\infty \le z'_1 = z_1 \le \cdots \le z'_{d_1-1} = z_{d_1-1} \le \xi_1 \le z'_{d_1} = z_{d_1+1} \le \cdots$$
$$\cdots \le z'_{d_2-1} = z_{d_2} \le \xi_2 \le z'_{d_2+1} = z_{d_2+1} \le \cdots \le z'_N = z_N \le +\infty\,. \qquad \text{(C10)}$$

Introducing

$$
\begin{aligned}
Z_{d_1,d_2}(\xi_1,\xi_2) &= Z_{d_1}(\xi_1) \cap Z_{d_2}(\xi_2)\,, \\
&= \{-\infty \le z_1 \le \cdots \le z_{d_1-1} \le \xi_1 \le z_{d_1+1} \le \cdots \le z_{d_2} \le \xi_2 \le z_{d_2+1} \le \cdots \le z_N \le +\infty\}\,, \qquad \text{(C11)}
\end{aligned}
$$

we find

$$
\begin{aligned}
A_B(\xi_1,\xi_2) =&\, c \sum_{d_1=1}^N \sum_{d_2=d_1}^N \sum_{\boldsymbol{y}\in Y(y_1=d)} \int_{Z_{d_1,d_2}(\xi_1,\xi_2)} \prod_{j=1,j\neq d_1}^N dz_j\, \bar{\chi}(z_1,\cdots,\underset{\underset{d_1}{\uparrow}}{\xi_1},\cdots,z_N;\boldsymbol{y})\chi(z'_1,\cdots,\underset{\underset{d_2}{\uparrow}}{\xi_2},\cdots,z'_N;\boldsymbol{y'})\,, \\
=&\, c \sum_{d_1=1}^N \sum_{d_2=d_1}^N \underbrace{\left[\sum_{\boldsymbol{y}\in Y(y_1=d_1)} (-1)^{\boldsymbol{y}+\boldsymbol{y'}} \underset{M}{\det}\left(e^{-iy_a\lambda_b}\right)\underset{M}{\det}\left(e^{iy'_a\lambda_b}\right)\right]}_{S_B(d_1,d_2)} \underbrace{\left[\int_{Z_{d_1,d_2}(\xi_1,\xi_2)} \prod_{j=1,j\neq d_1}^N dz_j\, \underset{N}{\det}\left[\bar{\phi}_{j_a}(z_b)\right]\underset{N}{\det}\left[\phi_{j_a}(z'_b)\right]\right]}_{I(d_1,d_2;\xi_1,\xi_2)}\,, \\
=&\, c \sum_{d_1=1}^N \sum_{d_2=d_1}^N S_B(d_1,d_2) I(d_1,d_2;\xi_1,\xi_2)\,. \qquad\qquad\qquad\qquad\qquad\qquad\qquad\qquad\qquad\qquad \text{(C12)}
\end{aligned}
$$

where $\boldsymbol{y'}$ depends on $\boldsymbol{y}$ via (C9) and $\boldsymbol{z'}$ satisfies the constraint (C10).

The fermionic integral is defined by

$$A_F(\xi_1,\xi_2) = \int_{\mathbb{R}^{N-1}} \prod_{j=1}^{N-1} dx_j\, \bar{\chi}(x_1,\cdots,x_{N-1},\xi_1)\chi(x_1,\cdots,x_{N-1},\xi_2)\,. \qquad \text{(C13)}$$

A similar analysis as above shows that the $\boldsymbol{z'}$ variables satisfy the same constraint as in the bosonic case (C10), however, now the connection between $\boldsymbol{y'}$ and $\boldsymbol{y'}$ is given by

$$
\begin{aligned}
y'_N &= d_2\,, \quad y_N = d_1\,, \\
y'_i &= y_i \quad \text{for } y_i < d_1\,, \\
y'_i &= y_i - 1 \quad \text{for } d_1 < y_i \le d_2\,, \\
y'_i &= y_i\,, \quad \text{for } d_2 < y_i\,.
\end{aligned}
$$
$$\text{(C14)}$$

Note that in this case the variables $y_1,\cdots,y_M$ cannot take the value $d_1$. Similar calculations as in the bosonic case give

$$A_F(\xi_1,\xi_2) = c \sum_{d_1=1}^N \sum_{d_2=d_1}^N \underbrace{\left[\sum_{\boldsymbol{y}\in Y(y_N=d_1)} (-1)^{\boldsymbol{y}+\boldsymbol{y'}} \underset{M}{\det}\left(e^{-iy_a\lambda_b}\right)\underset{M}{\det}\left(e^{iy'_a\lambda_b}\right)\right]}_{S_B(d_1,d_2)} \underbrace{\left[\int_{Z_{d_1,d_2}(\xi_1,\xi_2)} \prod_{j=1,j\neq d_1}^N dz_j\, \underset{N}{\det}\left[\bar{\phi}_{j_a}(z_b)\right]\underset{N}{\det}\left[\phi_{j_a}(z'_b)\right]\right]}_{I(d_1,d_2;\xi_1,\xi_2)}\,,$$

$$=c \sum_{d_1=1}^{N} \sum_{d_2=d_1}^{N} S_F(d_1, d_2) I(d_1, d_2; \xi_1, \xi_2) \,. \tag{C15}$$

with $\boldsymbol{z}'$ and $\boldsymbol{y}'$ satisfying the constraints (C10) and (C14). The charge functions are the same in the bosonic and fermionic case but the pseudo-spin functions are different.

### Appendix D: Evaluation of $S_B(d,d)$ and $S_F(d,d)$

When $d_1 = d_2 = d$ the pseudo-spin bosonic function (23) is given by

$$S_B(d,d) = \sum_{\boldsymbol{y} \in Y(y_1=d)} (-1)^{\boldsymbol{y}+\boldsymbol{y}'} \det_M \left( e^{-iy_a \lambda_b} \right) \det_M \left( e^{iy_a' \lambda_b} \right) \,, \tag{D1}$$

with $\boldsymbol{y} = \boldsymbol{y}'$. Because the dependence on $y_{M+1}, \cdots, y_N$ is encoded in the sign factor we have

$$
\begin{aligned}
S_B(d,d) =& (N-M)! \sum_{y_2=1}^{N} \cdots \sum_{y_M=1}^{N} \left( \sum_{P \in S_M} (-1)^P \prod_{j=1}^{M} e^{-iy_j \lambda_{P_j}} \right) \left( \sum_{Q \in S_M} (-1)^Q \prod_{j=1}^{M} e^{iy_j \lambda_{Q_j}} \right) \,, \\
=& (N-M)! \sum_{P \in S_M} \sum_{Q \in S_M} (-1)^{P+Q} e^{-iy_1(\lambda_{P_1}-\lambda_{Q_1})} \prod_{j=2}^{M} \left( \sum_{y=1}^{N} e^{-iy(\lambda_{P_j}-\lambda_{Q_j})} \right) \,, \\
=& \sum_{P \in S_M} \sum_{Q \in S_M} (-1)^{P+Q} e^{-iy_1(\lambda_{P_1}-\lambda_{Q_1})} \prod_{j=2}^{M} \left( N \delta_{\lambda_{P_j}, \lambda_{Q_j}} \right) \,, \\
=& (N-M)! M! N^{M-1} \,, \tag{D2}
\end{aligned}
$$

where in the second line we have used that $e^{i\lambda_{P_j} N} = e^{i\lambda_{Q_j} N} = 1$ after the summation of the geometrical progression.

The pseudo-spin fermionic function is

$$S_F(d,d) = \sum_{\boldsymbol{y} \in Y(y_N=d)} (-1)^{\boldsymbol{y}+\boldsymbol{y}'} \det_M \left( e^{-iy_a \lambda_b} \right) \det_M \left( e^{iy_a' \lambda_b} \right) \,, \tag{D3}$$

with $\boldsymbol{y} = \boldsymbol{y}'$ but in this case $y_1, \cdots, y_M$ cannot take the value $d$. Similar to the bosonic case we have

$$
\begin{aligned}
S_F(d,d) =& (N-M-1)! \sum_{P \in S_M} \sum_{Q \in S_M} (-1)^{P+Q} \prod_{j=1}^{M} \left( \sum_{y=1, y \neq d}^{N} e^{-iy(\lambda_{P_j}-\lambda_{Q_j})} \right) \,, \\
=& (N-M-1)! \sum_{P \in S_M} \sum_{R \in S_M} (-1)^{R} \prod_{j=1}^{M} \left( \sum_{y=1, y \neq d}^{N} e^{-iy(\lambda_{P_j}-\lambda_{PR_j})} \right) \,, \\
=& (N-M-1)! \sum_{P \in S_M} \sum_{R \in S_M} (-1)^{R} \prod_{j=1}^{M} F(P_j, PR_j) \,, \tag{D4}
\end{aligned}
$$

with $F(P_j, PR_j) = N \left[ \delta_{\lambda_{P_j}, \lambda_{PR_j}} - e^{-id(\lambda_{P_j}-\lambda_{PR_j})}/N \right]$. It is easy to see that the $N$-by-$N$ matrix with elements $e^{-id(\lambda_j - \lambda_k)}/N$ where $\lambda_j = \frac{2\pi i}{N} j$ with $j \in \{0, \cdots, N-1\}$ is of rank 1. Then, using von Koch's formula (valid for any $M$-by-$M$ matrix S)

$$\det_M(1 - \gamma S) = 1 + \sum_{n=1}^{M} \frac{(-\gamma^n)}{n!} \sum_{k_1, \cdots, k_n=1}^{M} \det \left( S_{k_p, k_q} \right)_{p,q=1}^{n} \,, \tag{D5}$$

for the determinant $\sum_{R \in S_M} (-1)^R \prod_{j=1}^{M} F(P_j, PR_j)$ we obtain $N^M (1 - \sum_{j=1}^{M} e^{-id(\lambda_{P_j}-\lambda_{P_j})}/N) = N^M(1 - M/N)$ due to the fact that only the $n=1$ terms in (D5) survive (the rest of the minors are zero). Therefore $S_F(d,d) = (N-M-1)! M! N^M (1 - M/N)$.

**Appendix E: One-point correlation function and momentum distribution of trapped free fermions**

For a system of $N$ harmonically trapped free fermions the wavefunction of the ground state is

$$\chi_{FF}(x_1, \cdots, x_N) = \frac{1}{\sqrt{N!}} \det_N [\phi_{a-1}(x_b)]_{a,b=1,\cdots,N} \,,$$

where $\phi_n(x) = c_n e^{-x^2/2l_{HO}^2} H_n(x/l_{HO})/l_{HO}^{1/2}$, $c_n = 1/\left(2^n n! \pi^{1/2}\right)^{1/2}$, $l_{HO} = (\hbar/m\omega)^{1/2}$ and the field-field correlator can be easily computed as

$$g_{FF}(\xi_1, \xi_2) = N \int \bar{\chi}_{FF}(\xi_1, x_2, \cdots, x_N) \chi_{FF}(\xi_2, x_2, \cdots, x_N) \, dx_2 \cdots dx_N \,,$$

$$g_{FF}(\xi_1, \xi_2) = \sum_{i=0}^{N-1} \bar{\phi}_i(\xi_1) \phi_i(\xi_2) \,. \tag{E1}$$

The density of particles in the trap is $\rho_{FF}(\xi) \equiv g_{FF}(\xi, \xi) = \sum_{i=0}^{N-1} \bar{\phi}_i(\xi) \phi_i(\xi)$. The momentum distribution $n_{FF}(p) = \int \int e^{ip(\xi_1-\xi_2)/\hbar} g_{FF}(\xi_1, \xi_2) \, d\xi_1 d\xi_2/(2\pi)$ can be computed with the help of the following formula which can be derived using Eq. 7.374(8) of [95]

$$\int_{-\infty}^{+\infty} e^{ikx} e^{-x^2/(2b^2)} H_n(ax) \, dx = i^n b \sqrt{2\pi} (2a^2 b^2 - 1)^{n/2} H_n \left[\frac{ab^2 k}{(2a^2 b^2 - 1)^{1/2}}\right] e^{-k^2 b^2/2} \,, \quad a, b \in \mathbb{R} \,. \tag{E2}$$

We find

$$n_{FF}(p) = l_{HO}^2 \sum_{i=0}^{N-1} \bar{\phi}_i \left(\frac{pl_{HO}^2}{\hbar}\right) \phi_i \left(\frac{pl_{HO}^2}{\hbar}\right) \,,$$

$$n_{FF}(p) = l_{HO}^2 \rho_{FF} \left(\frac{pl_{HO}^2}{\hbar}\right) \,, \tag{E3}$$

which shows that the momentum distribution of a system of trapped free fermions is proportional with its real space density. In units of $\hbar = m = \omega = 1$ this relation becomes $n_{FF}(p) = \rho_{FF}(p)$.

---

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
