# Peer review of "Dynamical fermionization in the one-dimensional Bose-Fermi mixture"

_SciPost Physics_

## Round 3 · Referee Report · Anonymous · 2021-10-17

Strengths

Based on recent developed methods to dynamical fermionization, the author studied this phenomenon in the 1D trapped Bose-Fermi mixture in strong coupling limit. The whole calculation based on the assumption that the wave function is fully decoupled from spin part and charge parts, see eq. Eq. (24). Then the calculation of the single particle reduced density matrix can be calculated in terms of determinants. I believe the obtained results are fine and mathematically correct.

Weaknesses

the presentation of the paper is awful. The section III are the main results, however, the logical derivations were briefly listed in Sections VII, and IX, even some of them ere given in Appendices. Such a scientific presentation make the reader frustrated in capturing the logical steps.

Report

The whole calculation is mathematically fine, but physical discussion is very poor. The dynamical ferminization has been well studied in the Lieb-Liniger gases, see [42] and
Malvania et al, Science 373, 1129 (2021);
Ka, et al, Science 371, 296 (2021).
In this Bose-Fermi mixture, there should have the same origin of such dynamical ferminization. The author did not give new aspects of such dynamical ferminization from spin degrees of freedom.

Requested changes

1) The section III are the main results, however, the logical derivations were briefly listed in Sections VII, and IX, even some of them ere given in Appendices. Such a scientific presentation make the reader frustrated in capturing the logical steps. I would suggest the authors fully rewrite the Section II, III and VII and IX as one section, and give physical description on wave function and the derivations of the ferminization correlation functions for the quench dynamics.
2) The key concern of mine is that why the wave function Eq. (24) is correct? How much does the spin part (spin part) affect the ferminization? The figures showed that the quench dynamics seemed to be determined by charge degree of freedom.
3) The description of the plotted figures in the main text lack enough description on their physics. The Sections IV, V, VI looks too trivial.
4) In the Section I, some new references are needed to update, for example,
Malvania et al, Science 373, 1129 (2021);
Ka, et al, Science 371, 296 (2021),
In particular, the generalized hydrodynamics of the 1D Bose-Fermi gases model (1) have been studied in Wang, et al, J. Phys. A, 53, 464002 (2020), etc.

  • validity: good
  • significance: good
  • originality: low
  • clarity: good
  • formatting: good
  • grammar: reasonable

Author:  Ovidiu Patu  on 2021-11-08  [id 1925]

(in reply to Report 1 on 2021-10-17)

List of general changes:

A)  In Sec. II we have presented additional arguments for our choice of wavefunctions Eq. (3) (suggested by Referee 1) and showed how in the limiting cases of a purely fermionic or purely bosonic system they reproduce the well known results for free fermions and TG bosons. (suggested by Referee 3).

B) Reorganized Secs. VII, VIII and XI (suggested by Referee 1). After the presentation of the analytical results for the correlation functions in Sec. III we present their derivation in Sec. IV.

C). Improved the physical discussion in Secs. IV, V and VI (now Secs. V, VI and VII) (suggested by Referees 1 and 2) and clarified and expanded the captions of the figures (suggested by Referee 3).

D) Expanded the Conclusions (Sec. VIII) (suggested by Referee 2) and elaborated on the extension of our  results in the case of strong but finite interaction (suggested by Referee 3).

Response to requested changes:

  1. Implemented the suggested changes (see A) and B) of General changes).

  2. We have presented additional arguments for the choice of the wavefunctions in Sec. II. Even though the dynamics is encoded in the charge degrees of freedom the pseudo-spin structure of the initial ground-state plays an important role influencing the momentum distribution of each component of the mixture during the time evolution.

  3. Implemented the suggested changes (see C) of General changes).

  4. Updated the reference list.

---

## Round 3 · Referee Report · Anonymous · 2021-10-27

Strengths

-Several analytical formulae for the correlation functions of the Bose-Fermi mixture have been reported. For each of these, a detailed derivation can be found in Sec. VIII-IX and in the Appendixes.

-The author considers a Luttinger liquid regime for the compound which is much less studied in literature than its spin incoherent counterpart. Furthermore, several results derived by the author have general validity.

Weaknesses

-The post quench dynamics is not dependent on the pseudo spin degrees of freedom. The discussion on the dynamical fermionization follows from that of a Tonks-Girardeau gas.

-The new features about the non equilibrium dynamics characterizing the mixture are poorly discussed, especially if compared to the details reported for the derivation of the analytical formulae for the initial state. In my opinion, in Sec. VI the differences with respect to the Tonks-Girardeau gas are highlighted but they lack of a physical interpretation.

Report

In this manuscript, the author considers the dynamics after a sudden release from harmonic confinement of a Bose-Fermi mixture in the limit of strong repulsion. For the trapped gas, the author writes down the eigenstates in the form of a determinant for the hard core particles times a pseudo spin function that takes care of the bosonic/fermionic nature of the compound (Sec. II and Appendixes A,B). Such exact eigenstates are then used to set the intial state of the system as the inhomogeneous ground state in the Luttinger liquid regime.
Using the determinant representation of the many body wavefunction and a useful parametrization introduced in Sec. VII, the author is able to show that the spin and the charge degrees of freedom can be factorized in the expression for correlation functions (Appendix C). After some calculations (detailed in Sec. VIII and IX), the author derives exact expressions for the one-particle density matrix and for the momentum distribution function, summarized in Sec. III.
The dynamics after a sudden trap release is discussed in Sec. IV-V. It is mainly based on known exact results for the Tonks-Girardeau gas released from harmonic confinement. Indeed, the pseudo spin degrees of freedom remain unchanged during the post-quench time evolution, allowing for a similar treatment of the dynamics.

Requested changes

1. I suggest to improve the discussion in Sec IV, V and VI. In my opinion, it is important to have a clear discussion of the dynamics of the charge degrees of freedom (which essentially follows from the results of the Tonks-Girardeau gas) and on the role of the pseudo spin functions (that remain frozen during the time evolution) from which a physical explanation of the observed differences between Tonks-Girardeau and the Bose-Fermi mixture can be presented, expanding on what is already written in Sec. VI.

2. I would suggest also to re-organize the contents of Sec. VII, VIII and IX and of the Appendixes. Although these contains non trivial exact calculations, I would try to summarize them in several appendixes. In the main text, I would write a small summary about the logic of these calculation at the end of Sec. III, before moving to the discussion of the dynamics.

3. The author can expand the section of conclusion. In the current version, this section provides even less information than those reported in the abstract.

4. The author can think about an analytical or numerical analysis of the same model but with a quench protocol that leads to a non trivial dynamics of the pseudo spin functions. In my opinion, this additional study will significantly raise the overall impact of the manuscript although it might be beyond the original scopes of this paper. I leave therefore the decision to the author.

  • validity: good
  • significance: good
  • originality: good
  • clarity: good
  • formatting: reasonable
  • grammar: good

Author:  Ovidiu Patu  on 2021-11-08  [id 1924]

(in reply to Report 2 on 2021-10-27)

List of general changes:

A)  In Sec. II we have presented additional arguments for our choice of wavefunctions Eq. (3) (suggested by Referee 1) and showed how in the limiting cases of a purely fermionic or purely bosonic system they reproduce the well known results for free fermions and TG bosons. (suggested by Referee 3).

B) Reorganized Secs. VII, VIII and XI (suggested by Referee 1). After the presentation of the analytical results for the correlation functions in Sec. III we present their derivation in Sec. IV.

C). Improved the physical discussion in Secs. IV, V and VI (now Secs. V, VI and VII) (suggested by Referees 1 and 2) and clarified and expanded the captions of the figures (suggested by Referee 3).

D) Expanded the Conclusions (Sec. VIII) (suggested by Referee 2) and elaborated on the extension of our  results in the case of strong but finite interaction (suggested by Referee 3).

Response to requested changes:

  1. Implemented the suggested changes (see C) of General changes)

  2. Reorganized Secs. VII, VIII and IX in one section after Sec. III. While the author is also of the opinion that the technical details can be relegated to the Apendices it has been strongly suggested that this presentation would frustrate a reader who wants to follow the derivations. Now the  paper has a more straightforward layout: results, their derivation followed by the discussion of the dynamics.

  3. Expanded the Conclusions.

  4. Two possible quench protocols not addressed are the periodic modulation of the trap frequency or a quantum Newton's cradle type scenario. It is possible that they would be addressed in a future publication in conjunction with considering other two-component models.

---

## Round 3 · Referee Report · Anonymous · 2021-10-28

Strengths

1. New analytic results on dynamical fermionization and correlation functions of one-dimensional Bose-Fermi mixture with strong interaction

2. Efficient numerical algorithm to evaluate correlation functions of one-dimensional Bose-Fermi mixture with strong interaction

3. Physical effects that could be potentially observed in cold atom experiments.

Weaknesses

The method used seems to be well established.

Report

A. Overview

The main result of the paper is about the dynamical fermionization in the one-dimensional strongly interacting Bose-Fermi mixture with analytic and numerical results on the explicit calculations of various correlation functions.

The author has done a great job explaining and documenting the technical details of analytic calculations, which is of great value for researchers in this field.

This paper has shown results that are of relevance for the ultracold atom experiments, which are possible to demonstrate the phenomena discovered in the paper.

B. Questions and suggestions

1. In the introduction, the author mentioned "In the TG (impenetrable) regime the ground state of such systems has a large degeneracy". However, from the Bethe Ansatz perspective, I would expect unique ground state with repulsive interaction, which can be reconciled with the "Bose-Fermi mapping". I'm wondering if there is any subtlety in my argument, and whether there is an intuitive way to understand the statement.

2. In Eq. (1), the interaction strengths of boson-boson and boson-fermion $g_{BB}$ and $g_{BF}$ defers by a factor of $\frac{1}{2}$ in the definition of the Hamiltonian. I presume that this parametrisation is natural to consider the integrability case when $g_{BB} = g_{BF} $. Is there any intuitive way to understand the factor of $\frac{1}{2}$ here?

3. Is there any intuitive way to understand the existence of the pseudo-spin degree of freedom for the bosons here? I think that when the fermion is absent, we do not expect the pseudo-spin degree of freedom to emerge. However, if I set the number of bosons $M=N$, the wavefunction (2) should reproduce the Tonks-Girardeaux wavefunction with no $\mathbf{\lambda}$. Is it the case here?

4. There is a "duality" between hard core bosons and non-interacting fermions, as explained recently in arXiv:2109.08626(https://arxiv.org/abs/2109.08626). I'm wondering if there's any "duality" that could relate the wavefunction with $M$ bosons and $(N-M)$ fermions to the one with $(N-M)$ bosons and $M$ fermions

5. About Fig. 3, I'm a little bit confused about the plot with solid and dashed lines. It would be great if the author could explain a little bit more the difference between the solid and dashed lines in the caption.

6. In the conclusion, the author mentioned that the features should hold in the case of strong but finite interaction strength. I'm wondering if the author could elaborate a bit more on this topic. I presume that for finite interaction strength, one might need to use the integrability to solve such problem. Just out of curiosity, I'm wondering if the integrability for the Bose-Fermi mixture is only available in the form of coordinate Bethe ansatz, or there is an algebraic description (quantum inverse scattering method). If so, should it be nested (related to higher rank Lie superalgebra)?

7. Some typos can be found in the Requested changes part below.

C. Conclusions

The paper is a nicely written article with lots of technical details. However, I think that it does not satisfy the Expectations part of the criterion of SciPost Physics (https://scipost.org/SciPostPhys/about#criteria). Instead, I find it perfectly matches with the criterion of SciPost Physics Core (https://scipost.org/SciPostPhysCore/about#criteria) with minor revision. I suggest the editor to consider the paper to be published in SciPost Physics Core with minor revision.

Requested changes

1. In the abstract, the last sentence "can also be implemented numerically ..." should be changed into "can be implemented numerically ...".

2. In the introduction part, "From the analytical point of view will prove that ..." should be changed into "From the analytical point of view ${ \it we}$ will prove that ...".

3. Title of Section III "Analitical formulae ..." should be changed into "${\it Analytical}$ formulae ...".

4. Other comments/changes can be found in the Report section.

  • validity: top
  • significance: good
  • originality: good
  • clarity: high
  • formatting: good
  • grammar: excellent

Author:  Ovidiu Patu  on 2021-11-08  [id 1923]

(in reply to Report 3 on 2021-10-28)

List of general changes:

A)  In Sec. II we have presented additional arguments for our choice of wavefunctions Eq. (3) (suggested by Referee 1) and showed how in the limiting cases of a purely fermionic or purely bosonic system they reproduce the well known results for free fermions and TG bosons. (suggested by Referee 3).

B) Reorganized Secs. VII, VIII and XI (suggested by Referee 1). After the presentation of the analytical results for the correlation functions in Sec. III we present their derivation in Sec. IV.

C) Improved the physical discussion in Secs. IV, V and VI (now Secs. V, VI and VII) (suggested by Referees 1 and 2) and clarified and expanded the captions of the figures (suggested by Referee 3).

D) Expanded the Conclusions (Sec. VIII) (suggested by Referee 2) and elaborated on the extension of our  results in the case of strong but finite interaction (suggested by Referee 3).

Answers to questions:

  1. At strong but finite interaction the Bethe ansatz indeed produces a unique ground-state. However, at $g=\infty$  the charge and spin degrees of freedom  decouple completely and the all the pseudo-spin states have the same energy. Adiabatically decreasing the strength of the interaction one of these states can be continuously connected with the unique ground-state at strong but finite coupling (in our case this is the state described by Eq. 9). Even in the  case of homogeneous multi-component systems (see for example [A.G. Izergin and A.G. Pronko, Nucl. Phys. B 520, 594 (1998)])  the spin-sector influences  the BAEs for the charge degrees of freedom only through a twist which is the total momentum of the spin-state. This also implies a large degeneracy due to the fact that there are many spin states with the same total momentum.

  2. If the strengths of the interactions between both types of particles are equal and given by g/2 then in the second quantized Hamiltonian  the contact interaction term for bosons is   $ (g/2)\psi_B^+\psi_B^+\psi_B\psi_B $ for  fermions is $ (g/2)\psi_F^+\psi_F^+\psi_F\psi_F $ (this is zero because polarized fermions do not "feel" the delta interaction) and the mixed terms   are $ (g/2)\psi_B^+\psi_F^+\psi_F\psi_B $ and $ (g/2)*\psi_F^+\psi_B^+\psi_B\psi_F $ (these are both equal). This is the origin of the 1/2 difference in the interaction terms of the Hamiltonian.

  3. The pseudo-spin sector describes the boson-fermion degrees of freedom and a pseudo-spin excitation would change the number of bosons and fermions by one keeping the total number of particles constant. When the system is purely bosonic (M=N) the wavefunction (3) of the main text  reduces to the TG gas wavefunction of Girardeau. We have modified Sec. II to include a discussion on the limiting cases M=0 and M=N.

  4. In the case of multi-component systems generalizations of the Bose-Fermi mapping are ill-advised in the opinion of this author (the main reasons being a) the resulting wavefunctions have too ``much symmetry" when exchanging particles of different types and b) lack of completeness in the spin space).  In the BFM case the wavefunction (3) which describes M bosons and N-M fermions can be transformed in a wavefunction for N-M bosons and M fermions  as follows: replace the Slater determinant for the charge degrees of freedom  with the TG wavefunction (multiply the det with $\prod_{a<b} sign(x_b-x_a)$) and have the integer n's describing the positions of the fermions instead of the bosons.

  5. Clarified in the text and caption of Fig. 3 the quantities described by the dashed lines. In Fig. 3 (a), (c) and (e) the dashed black lines represent the FWHM dynamics of the momentum distribution of M Tonks-Girardeau bosons subjected to the same confinement quench (sudden change of the trap frequency). In Fig. 3 (b), (d) and (f) the  dashed black lines represent the FWHM dynamics of the momentum distribution of N − M free fermions undergoing the  same quench.

  6. a) In Sec. VIII (Conclusions) we have elaborated on the reasons why some of the features of DF should hold in the case of strong but finite interactions following [Phys. Rev. Lett. bf 127, 023002 (2021)]. This is due to the fact that (to leading order) the wavefunction remains factorized and the spin sector is described by an XXZ spin-chain with variable exchange coefficients. What is interesting is that in the case of release from a trap the time-evolution of the spin Hamiltonian is given by a simple scaling with the solution of the Ermakov-Pinney equation which means that the spin configuration remains frozen also in this case and then we can apply the same logic as in Sec. VI. b) The answer is yes to both questions. Probably the most general solution for a mixed multi-component system of NLS type in the   graded QISM  framework is [Yu-Kui Zhou, Nuclear Physics B326, 775-786 (1989)]. For a system with two types of fermions and one type of bosons see [Z.-X. Hu, Q.-L. Zhang, and Y.-Q. Li, J. Phys. A: Math. Gen. 39, 351 (2006)].

  7. Corrected the typos.

We respectfully disagree with the assessment that the paper is not suitable for publication in SciPost Physics. In the first part of Sec. VIII (Conclusions) we have elaborated on the relevance and importance of our work presenting arguments which we hope would help in changing the initial evaluation of the referee.

---

## Editorial Decision

editor-in-charge_assigned